



# Late-Quaternary hydrological evolution of Fuente de Piedra playa-lake (southern Iberia) controlled by neotectonics and climate changes

Alejandro Jiménez-Bonilla[1], Lucía Martegani[2,3], Miguel Rodríguez-Rodríguez[1], Fernando Gázquez[2,3], Manuel Díaz-Azpíroz[1], Sergio Martos[4], Klaus Reicherter[5], Inmaculada Expósito[1]

[1]Department of Physics, Chemistry and Natural Systems, University Pablo de Olavide, Seville, 41013, Spain
[2]Water Resources and Environmental Geology Research Group, Department of Biology and Geology, University of Almería, Almería, 04120, Spain
[3]Andalusian Centre for the Global Change - Hermelindo Casto. ENGLOBA. University of Almería, Almería 04120, Spain.
[4]Geologicla Survey of Spain (IGME), Madrid, 28003, Spain
[5]Institute of Neotectonics and Natural Hazards, RWTH Aachen University, 52062, Germany

*Correspondence to*: Alejandro Jiménez-Bonilla (ajimbon@upo.es)

**Abstract.** Playa-lakes developed in semi-arid regions are sensitive to water input reductions, which may be influenced not only by climate changes and human management but also by changes in the size of the watershed. We accomplished an

interdisciplinary study combining structural, geomorphic, sedimentological, mineralogical and hydrological analyses to better understand the Fuente de Piedra (FdP) playa-lake evolution in southern Spain. By using previously published temperature and precipitation reconstructions, we assess the potential evapotranspiration and the runoff to estimate the maximum lake level during the FdP lifespan (>35 ka). Our results indicate that the FdP playa-lake level never exceeded 5m, although deposits at its NE margin are up to 15 m above the current lakebed at present. These lacustrine deposits are slightly tilted towards the SW.

The electrical conductivity profiles of groundwater in the FdP's shore and surroundings reveal a more pronounced interface between brackish water and brine in the northern part compared to the southern part of the basin. This implies that saline water once occupied the northern playa-lake margin in an area that is hardly ever flooded at present. The presence of reworked gypsum in the sedimentary sequence of the southern margin (down to a depth of 14 m), indicates substantial erosion of prior gypsum deposits, possibly redistributed from northern deposition areas. Altogether, our data suggest a SW displacement of the

playa-lake depocenter caused by an uplift of the NE and subsidence of the SW area. This shift is congruent with the combined effect of both the La Nava sinistral-normal fault and the Las Latas dextral-normal fault, respectively at the E and S margins of the FdP playa-lake.

## 1 Introduction

Saline playa-lakes develop in semi-arid to arid regions, where precipitation is considerably lower than evapotranspiration, leading to negative water balances. Water inputs are direct precipitation and runoff from the catchment, while water outputs





are evapotranspiration and infiltration. Slight changes in the water balance led to significant seasonal and interannual water level variability.

Under negative water balance conditions, the average flooded surface shrinks, eventually giving rise to complete desiccation
and playa-lake disappearance (Moral et al., 2013). Several factors control the change in the water discharge to playa-lakes, such as groundwater extractions for agricultural purposes (Rodríguez-Rodríguez et al., 2015) or climate changes (Schöder et al., 2018; García-Alix et al., 2022). In this regard, variations on temperature and/or precipitation have happened naturally over the Quaternary (e.g., Mann, 2002; Hughes et al., 2013), resulting in changes in evapotranspiration and lake inputs. Besides, recent studies, which modelled the vertical lake evolution, suggest that anthropogenic climate change during the Late Holocene
may have provoked additional changes in lake water input and evapotranspiration (e.g. Matthews, 2008). The bathymetry of these wetlands is characterized by planar surfaces and they show low sedimentation rates (e.g. García-Alix et al., 2022). In general, they are ephemeral water bodies, where evaporation leads to the formation of salt crusts during the dry season (e.g. Sánchez-Moral et al., 2002). When these wetlands are dry, they are sensitive to aeolian erosion because of the formation of clay-salt aggregates (Moral, 2016).

Climate changes are recorded in the sedimentary playa-lake register, so playa-lake sediments can provide valuable paleoclimate information (Roberts et al., 2001; Höbig et al., 2016; Schöder et al., 2018, 2020; Cohen et al., 2022; García-Alix et al., 2022, among others). However, accurate interpretations of their sedimentary records require a deep knowledge of the landscape evolution in the surrounding areas, especially those changes related to variations in their catchment size and geometry. Although these changes in both flooded areas and watersheds of playa-lakes have been poorly investigated to date,
recent studies have demonstrated that such water bodies are quite dynamic, especially those settled in active orogens (e.g. Berry et al., 2019), where tectonic processes may favour divide migration and watershed capture, leading to variations on the watershed area (Jiménez-Bonilla et al., 2023a).

The Fuente de Piedra (FdP) playa-lake is the largest playa-lake within an endorheic area located at the Atlantic-Mediterranean water divide in southern Spain (Figs.1 and 2). The FdP playa-lake remains completely desiccated during the dry season (May
to October), except for exceptionally wet interannual periods, when the lake level can reach more than 2m (Rodríguez-Rodríguez et al., 2016). The FdP watershed developed in a depressed area within the Betics fold-and-thrust belt, which is limited by topographic highs controlled by transpressional shear zones with evidence of moderate Quaternary tectonic activity (Jiménez-Bonilla et al., 2016; 2023a; 2023b). In the past decades, several studies have assessed the modern hydrology and paleohydrological evolution of this endorheic system (e.g. Kohfahl et al., 2008; Höbig et al., 2016; Rodríguez-Rodríguez et
al., 2016). These investigations assumed that the FdP playa-lake surface as well as its watershed geometry and extension have remained constant since its origin (at least 35 kyrs BP; Höbig et al., 2016). Consequently, variations in the lake flooded surface had been attributed to paleoclimate changes. Indeed, presumed Pleistocene lake terraces (10 m above the current lake floor) were reported and interpreted as evidence for lake highstands during humid periods (Höbig et al., 2016).

Here, we develop an evolution model for the FdP playa-lake based on structural, hydrological and sedimentological
information. The general aims of this paper are i) to re-examine the role of late Quaternary tectonic activity in the FdP playa-





lake, ii) to propose an integrated chronology of FdP playa-lake sediments after careful characterization of its sediments and tectonic setting and iii) to investigate potential impact of climate changes on the lake by modelling the water level over the past 35 kyrs upon both lithological and mineralogical variations in the FdP sedimentary sequence and changes in the geometry of the flooded area.

## 70  2 Geological setting

The Betic Cordillera, which is the northern branch of the Gibraltar arc, is built up by the Neogene collision between the Alboran domain (hinterland) and the South Iberian paleomargin (Vera, 2004; Fig. 1A). During the westward migration of the Alboran domain, Flysch units were deposited and later sandwiched between both domains. The South Iberian paleomargin was deformed into the Betics fold-and-thrust belt, which registers a main, lower to middle Miocene deformation and a late post-

Serravallian deformation event, responsible for the main current relief. This late deformation is still active along the belt (e.g., Balanyá et al., 2012; Jiménez-Bonilla et al., 2015).

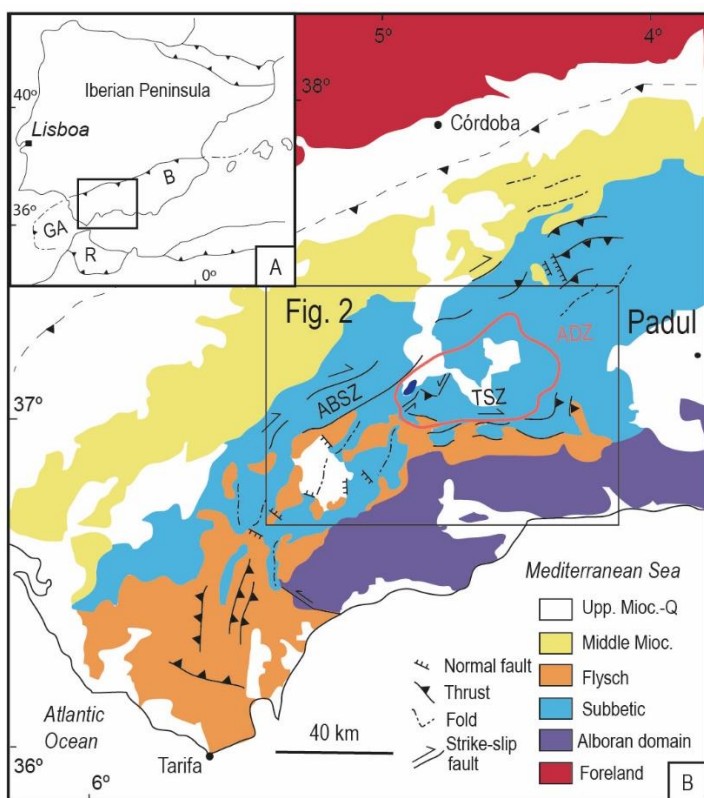





**Figure 1: (a) Inset location map. (b) Tectonic map of the western Betics showing the Atlantic-Mediterranean water divide and placing the Padul peatland, used here for water level calculations (see section 3.1). TSZ: Torcal Shear Zone; ABSZ: Algodonales-Badolatosa Shear Zone; ADZ: Antequera Depressed Zone.**

## 2.2 The Antequera Depressed Zone (ADZ)

The study area is the FdP endorheic watershed, which lies in an E-W elongated depressed area in the Betics fold-and-thrust belt. This basinal area, hereafter referred to as the Antequera depressed Zone (ADZ, Figs. 1 and 2), is limited by mountain ranges which are mainly controlled by two post-Serravallian, dextral transpressive zones: the Algodonales-Badolatosa Shear Zone (ABSZ; Jiménez-Bonilla et al., 2015) to the NW, and the Torcal Shear Zone (TSZ, Díaz-Azpiroz et al., 2014; Barcos et al., 2015) to the S and SW (Figs. 1 and 2). Both the ABSZ and TSZ show evidence of upper Miocene to recent tectonic activity

(Barcos et al., 2015; Díaz-Azpiroz et al., 2020; Jiménez-Bonilla et al., 2023). Both shear zones produce topographic highs located at 600-800 and > 1,000 m a.s.l. for the ABSZ and TSZ, respectively (Fig. 2), which sharply drops to the ADZ. The ADZ is characterized by a mostly flat topography at approx. 450 m a.s.l. It includes some scattered reliefs, as the Humilladero range (Fig. 2), generated by a compressive bridge that links two left-lateral shear zones of the post-Tortonian Humilladero Transverse Zone (HTZ, Fig. 2). Some studies suggest that the ADZ nucleated as an endorheic area since the emersion of this

Betics segment in Pliocene to Pleistocene times (Elez et al., 2018; 2020). The ADZ was later captured and partially drained by Guadalhorce and Guadalteba rivers to the Mediterranean and by the Guadalquivir network to the Atlantic (Figs. 1 and 2). Thus, the main Quaternary Atlantic-Mediterranean water divide is located in the western Betics, but it is not a sharp feature. In fact, it is a km-wide diffuse area presenting several endorheic basins (Rodríguez-Rodríguez et al., 2010): (1) the FdP watershed, which includes the FdP and La Nava playa-lakes, (2) the Campillos (C) endorheic basin, made up by more than 10

playa-lakes, (3) the Archidona (A) playa-lakes, (4) the Ratosa (R) endorheic basin and (5) Lomas (L) playa-lakes (Fig. 2).



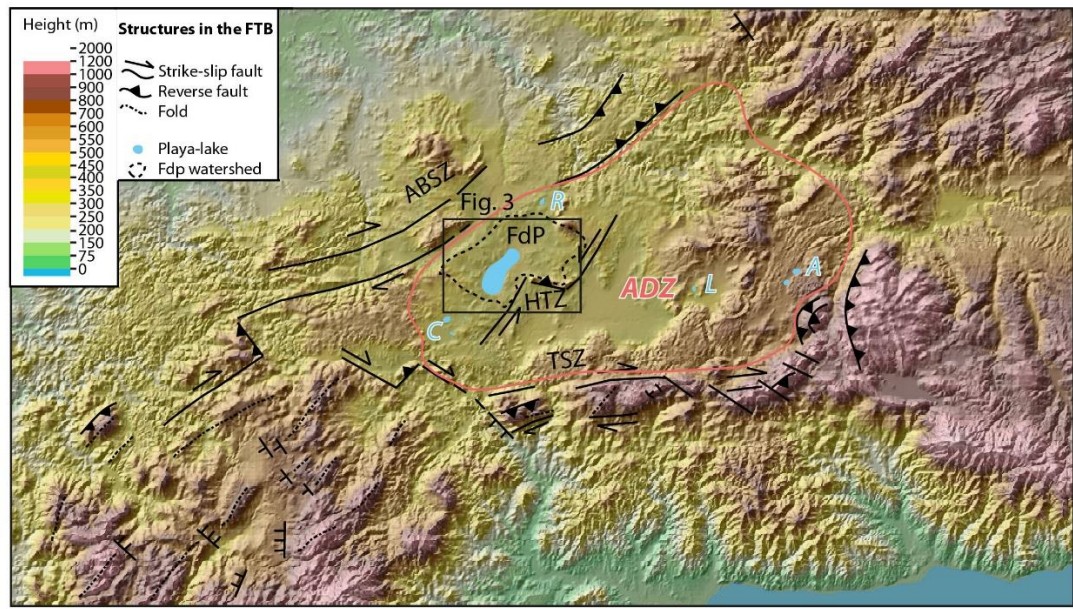

**Figure 2: Topographic map and hillshade of the Antequera Depressed Zone (ADZ) showing the main playa-lakes: FdP (Fuente de Piedra), R (Ratosa), C (Campillos), L (Lomas) and A (Archidona), and the main structures grouped into three main shear zones: TSZ (Torcal Shear Zone), ABSZ (Algodonales-Badolatosa Shear Zone) and HTZ (Humilladero Transverse Zone).**

### 2.3 FdP watershed rocks

The lithological formations that crop out in the FdP watershed (Figs. 1 and 2) belong to the Subbetic units. They derive from the most internal position of the South Iberian paleo margin and are made up of Tortonian to Messinian shallow-marine deposits and Pleistocene to Holocene continental deposits (Vera et al., 2000). Subbetic units are composed of: (1) Triassic clays, gypsum and marls that include isolated dam- to hm-scale dolostone bodies; (2) Jurassic dolostones and limestones that constitute the main mountain ranges (Sierra de los Caballos in the ABSZ and Sierra de Humilladero in the HTZ; Fig. 3) and (3) Cretaceous to Paleogene marls and marly-limestones. Recent works have interpreted the Triassic rocks outcropping in this area as an allochthonous canopy and the main mountain ranges made up of Jurassic limestones as tectonic windows (Flinch and Soto, 2017; 2022). Shallow-marine deposits testify to the progressive shallowing of the ADZ. They are Tortonian to Messinian calcarenites and marls (Cruz-Sanjulián, 1991), probably deposited in small-size basins (Flinch and Soto, 2017), that are currently deformed and often lie unconformably over Subbetic units. Pliocene sediments vary from polygenic sandstones associated with beach-like environments to continental conglomerates related to alluvial fans. The source of these conglomerates and sandstones is located immediately to the south of the TSZ and includes Alborán domain-related clasts of schists, marbles and gneisses, together with sandstones and limestones clasts coming from Flysch and Subbetic units.

Pleistocene to Holocene sedimentary sequences are stream-related sediments such as alluvial terraces, alluvial fans and playa-lake-related deposits (e.g., sandy aeolian lunettes and other lacustrine deposits, Fig. 3).





125

**Figure 3: (a) Geological map of the FdP playa-lake and its watershed (LBL: lake base level) with the location of the two drilling cores used in this work. Topographic cross-sections showing (b) the FdP sediments slope, (c) the geometry of an erosive pediment to**





the W of the FdP playa-lake, (d) the geometry of an erosive pediment to the E of the FdP playa-lake, (e) the sinistral-normal, La Nava Fault, which separates the FdP and La Nava (N) playa-lake and (F) picture of a left-lateral strike-slip fault associated with La Nava fault.

### 2.4 Hydrological features of the FdP playa-lake

The FdP playa-lake occupies a conspicuously eccentric location at the SW edge of a topographically closed watershed of about 150 km². The flat surface - flooded seasonally - has an NE-SW elongated ellipsoidal shape (6.8 km long and 2.5 km wide), which it is, in turn, aligned with the Ratosa playa-lake to the NE and the Campillos playa-lakes to the SW of the FdP watershed. (Fig. 2). The FdP playa-lake is defined as a hyper saline lake of the Cl-(SO₄)-Na-(Mg)-(Ca) type. Concentration of dissolved solids range from 18 to 200 g/l in the surface water (Rodríguez-Rodríguez, 2002). The FdP playa-lake has a lifespan of at least 26 kyrs and probably longer than 45 kyrs (Höbig et al., 2016). Under natural conditions, the FdP playa-lake receives surface and subsurface runoff from its watershed. The outputs correspond only to evapotranspiration (Rodriguez-Rodriguez et al., 2016). In summer, the FdP is completely dry, but minimum piezometric levels under the playa floor ranges from -20 to -30 cm (Kohfahl et al., 2008). This is consistent with a discharge-type playa-lake with a hydrogeological basin that is almost coincident with the surface watershed (Kohfahl et al., 2008). In the past, the playa-lake was the lowest part of the hydrogeological basin, and received water inputs also from the N and NE sectors from a spring located in the foothills of the carbonate aquifer of Mollina mountain range (by means of the Santillán stream; Fig. 3) and, most probably, groundwater inputs also from the Sierra de Humilladero mountain range (Figs. 2 and 3), although no springs have been observed in this carbonate aquifer. Both aquifers are intensively exploited at present (e.g., Rodríguez-Rodríguez et al., 2015). Below the playa-lake, a hypersaline stable groundwater brine has been developed via evapotranspiration and convection processes (Kohfahl et al., 2008; Benavente et al., 1993). The high salinity and density of the water in the playa-lake surface favour the slow percolation of this brine below the lake. This brine is detached from the fresh groundwater of the aquifer by a freshwater – saltwater interface, similar to that of coastal aquifers, being the playa-lake perimeter the equivalent to the coastal shore. A summary of relevant studies about the hydrogeology of the FdP playa-lake can be found in Rodríguez-Rodríguez et al. (2016).

### 3 Methodology

### 3.1 Hydrological data compilation and water level calculation

We use hydrological information of FdP from several sources (e.g. scientific reports from the Geological Survey of Spain and the Andalusian Autonomous Government) to elaborate the hydrograph of the daily evolution of the water level in the playa-lake. The water level above and below the lake floor was obtained from an automatic water level sensor installed in a piezometer placed inside the lake basin. Depth was expressed as altitude above the sea level (m a.s.l.).



We reconstructed the paleo-water level of the FdP playa-lake to compare with lake sediments. To do that, we conducted water
balances every 50 yrs over the last 35 kyrs (700 water balances). We applied a lake water balance equation developed in
previous studies of southern Iberian playa-lakes (e.g. Rodríguez-Rodríguez et al., 2015). The lake water balance considers
average climatic conditions and is expressed as:

$$\Delta V = P–E + Si−So + GD–GR, \qquad (1)$$


where $\Delta V$ (mm) is the change in the lake water level, whose bathymetry is completely flat at 410 m a.s.l., P (mm) stands for
precipitation onto the lake surface, and E (mm) is the direct evaporation from the flooded surface. Si stands for the surface
runoff, which is calculated by means of a soil water balance. So is the surface outflow that, in the case of FdP, is set to be 0.
GD represents the groundwater discharge from the regional aquifer to the lake and GR is the groundwater recharge from the
lake to the aquifer. We assume that Si + GD - GR equals BD (Basin Discharge) i.e., the total volume of water runoff, surface
and groundwater, flowing into the playa-lake from the watershed. In the case of FdP, previous studies suggest that the limit of
the surface watershed coincides with that of the groundwater watershed (e.g., Linares, 1990), so BD can be calculated easily.
Consequently, eq. 1 is simplified to:

$$\Delta V = P–E + BD \qquad (2)$$

Mean precipitation data (mm/year) were obtained from the sedimentary record of Padul peatland (See Fig. 1B for location), at
about 80 km from FdP to E, which is based on fossil pollen data (Camuera et al., 2022). There are no climatic differences
between both locations. We filled data gaps by interpolation using linear regression between available data (Fig. 4).
Atmospheric temperature data from 0 to 15 kyrs BP were obtained from Sea Surface Temperature (SST) reconstructions in
the Alboran Sea by Català et al. (2019) and by Morcillo-Montalbá et al. (2021) from 15 to 35 kyrs BP. We assume that the
long-term air temperature in the surroundings of FdP (~50 km from the seashore) varied in the same fashion and magnitude as
SST. We used the paleo-temperature record to calculate potential evapotranspiration (PET) by the Thornthwaite method, using
TRASERO software for the analysis and calculation of hydrological time series (Rodríguez-Hernández et al. 2007). We
corrected these values by multiplying by 1.15 because this method usually underestimates real evapotranspiration (e.g.
Rodríguez-Rodríguez et al., 2016). We obtained monthly potential E values that were later converted to annual potential E
values. BD, obtained from the soil water balance, was calculated with the same software using monthly potential E and the
water holding capacity (WHC) (Rodríguez-Hernandéz et al., 2007). We considered that water inputs only come from the
watershed, which is assumed to have remained constant over the past 35 kyrs. We ran the calculations (Allen et al., 2011)
using two values of WHC, 50 and 75 mm, to create two series of maximum lake level. We have applied low permeability
values according to the rocks cropping out in the FdP watershed. We converted monthly water surplus to yearly surplus and





then multiplied this value by the FdP watershed / average flooded area relationship (W/AFS) to obtain the amount of yearly basin discharge (BD) in mm. We assume that when the water level is between 0 and 2 m, the AFS remains constant, so that the W/AFS relationship does not change (Rodríguez-Rodríguez et al., 2016). Then, we obtained the increase on the lake level

(ΔV) for every 50 yrs. Considering the FdP playa-lake behaviour from 1983 to 2012, we also assume that the FdP water level must start at 0 mm every year in most cases, then ΔV equals to lake level. Importantly, to obtain the maximum water level we must sum the ΔV to the water level at the end of the previous hydrological year during wet years. We cannot make calculations for every year because the step of our paleoclimate series is 50 years (Fig. 4). Hence, we chose the most humid period (P = 700 mm, ETo = 850 mm and runoff = 75 mm; Fig. 4) and made iterative calculations per year during 50 yrs, holding climate

conditions the same, but changing the W/AFS relationship. Thus, we recalculated the AFS for every year depending on the water level reached in the previous year.

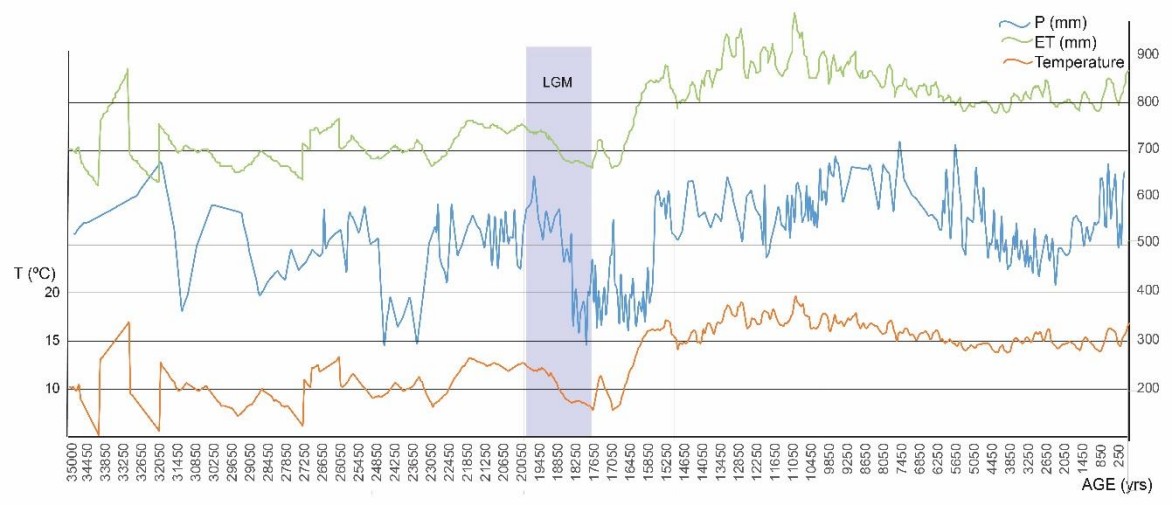

**Figure 4: Temperature, ET and precipitation reconstruction during the past 35,000 years in southern Iberia (Català et al., 2019; Morcillo-Montalbá et al., 2020; Camuera et al., 2022).**


## 3.2 Geomorphological and structural data

We improved the previous cartography of the FdP playa-lake by focusing on the recent erosive and depositional landforms. Using slope maps, ortho imagery and a 1:10,000 digital elevation model (Junta de Andalucía, 2002), we made an accurate cartography of playa-lake sediments associated with the FdP. We also constrained the Pliocene to Pleistocene landforms (Fig.

3). We collected structural data from those structures that affect Pliocene to younger sediments, whose scale ranges from several 100 m to kilometric-scale (Fig. 3).





**3.3 Compilation of geochronological and sedimentological data of the FdP sediments**

We combined previously published geochronological data of FdP sediments to create a chronological framework for the lake
evolution and discuss the reliability of the current age model. In this regard, Höbig et al. (2016) obtained seventeen AMS
radiocarbon ages, conducted on bulk sediment, pollen, charcoal, remains of plants and seeds from the 14-m-long sediment
core 2013-04, recovered from the southern margin of the lake (table 1, Fig. 5). In addition, we have used eight recent U-Th
ages obtained from gypsum crystals of core 2012-PL1 (1.4 m long) collected from the central part of the basin (Obert et al.,
2022). Radiocarbon dates were calibrated into calendar years using the IntCal 13 curve (Reimer et al., 2013). A Bayesian age-
depth model was calculated using the "Rbacon" R-based package (version 2.5 - March 2023, Blaauw and Christen, 2011) (Fig.
5).

We investigate in detail the sedimentology of cores 2013-04 and 2012-PL1, which mostly comprises laminated clayey and
carbonated sediments as well as different types of gypsum deposits ranging from gypsum sand to several cm-long prismatic
selenite crystals. We primarily focus on the morphology of the gypsum crystals to evaluate either the primary or diagenetic
origin of gypsum at various depths and use the criteria of Cody and Cody (1988) to identify different depositional environments
between the FdP southern margin and its central part.

## 4 Results

### 4.1 Geology of the FdP watershed

In this section, we describe the morphology of sedimentary bodies originated during the FdP development (Pleistocene to
Holocene) and the recent structures that have been active at least during the last 2 Myrs and may influence the FdP watershed
and its flooded area. area.

#### 4.1.1 Geomorphology of Pleistocene to Holocene sedimentary bodies

We differentiate sediments from two young depositional environments: fine-grained partly evaporitic, lacustrine deposits
related to the FdP playa-lake and colluvial Pleistocene and Holocene pediments.
Lacustrine sediments related to the current FdP playa-lake crop out quasi-horizontally at 410 m a.s.l. (lake base level, LBL)
and extend to the NE, out of the playa-lake current area, where they reach altitudes of 425 m a.s.l. (Fig. 3). Here, they show a
smooth slope (from 1 to 1.5%) towards the SW (Fig. 3B). Towards the E, lacustrine deposits of the La Nava playa-lake are
currently isolated from the FdP playa-lake by a topographic high. Here, they are horizontal and are located 3 m above the FdP
LBL (respective altitudes of 413 m *vs* 410 m; Figs. 3A and E).
The FdP watershed presents both depositional and erosive pediments. They are characterized by red soils, sometimes called
"red rendzinas", "terra rossa" or chromic luvisols, developed over Miocene limestones, which are very abundant in the
Mediterranean (Sandler et al., 2015). Occasionally, these soils also include carbonate crusts, so-called calcretes, which
typically form in Mediterranean climate (Alonso-Zarza et al., 2010). The most intensive development of such red



Mediterranean soils occurred from the Miocene to the Late Pleistocene, due to the numerous climate fluctuations in those
periods (Atalai, 2002). Pleistocene ages seem to be consistent with our observations, as red soils overlie Pliocene to Pleistocene
conglomerates and sandstones, and are covered by Holocene lacustrine deposits (Fig. 3A). Their thickness varies from 0 to 0.5
m in the case of erosive pediments whilst they reach more than 1 m in the case of colluvial pediments (glacis deposits). They
all constitute flat surfaces that are situated about 10 to 70 m above the current FdP LBL (Figs. 3A, 3C and 3D). The best-
preserved one is located to the W of the FdP playa-lake, at about 470 m a.s.l. (Fig. 3C). It is related to an erosive pediment that
truncates Tortonian to Messinian calcarenite beds (Fig.3C) and extends over 1,000 hm$^2$. Headwater erosion from both Atlantic
rivers and FdP streams are eroding this thin red soil (Fig. 3A). The thickest depositional pediments are found in depressed
areas (at ca. 420 m a.s.l.), close to the current FdP LBL (Fig. 3A). The red rendzinas soils in these deposits have a complete
soil profile.

### 4.1.2 Tectonic structures and kinematics

The structures involving Tortonian to Holocene sediments are faults that can be grouped into three sets:
The NE-SW striking, La Nava fault zone, related to the western branch of the HTZ, affects Tortonian to Messinian sediments
SE of the FdP playa-lake (Figs. 3A and 3F). This fault is approx. 10 km long and extends to the S of the Campillos playa-lake
system. Fault surfaces strike N20-45ºE and dip between 70-90º to the WNW. Slickenlines show pitches up to 15º to the SSW
(Fig. 3F). This, combined with kinematic indicators such as S-C-like structures and calcite and gypsum slickenfibres, indicates
a dominant left-lateral strike-slip movement and a subordinate normal movement throwing the western block down (Figs. 3A
and 3F). Other highly dipping NE-SW faults affect Jurassic and Pliocene outcrops to the W of the FdP playa-lake (Fig. 3A).
WNW-ESE to NW-SE trending right-lateral faults constitute the 10 km long, Las Latas fault zone, that coincides with the
southern water divide of the FdP watershed. Fault surfaces strike N60-85ºE and dip 65-90º towards the NNE. Slickenlines
show pitches up to 35º to the ESE. Kinematic indicators such as S-C-like structures, indicates a dominant right-lateral strike-
slip movement with a subordinate normal component, which lowers the northern block where the FdP watershed is located
(Fig. 3A). We interpret this fault zone as a synthetic Riedel fault of the ABSZ (Figs. 2 and 3A). Other WNW-ESE faults are
found in the W margin of the FdP watershed (Fig. 3A).
The western water divide of the FdP watershed coincides with the hinge zone of a kilometre-scale antiform, deforming
Tortonian to Messinian units. Its axis plunges. 25º towards the SW and its hinge zone is locally truncated by a Pleistocene
erosive pediment (Figs. 3A and 3C).

## 4.2 Chronological framework and description of sedimentary facies

The radiocarbon dates of the sediments in core 2013-04, from the southern margin of the lake, range from 0.8 to 45 kyr BP
(Table 1) (Höbig et al., 2016). They show large dispersion and age reversals, especially in the lower section, which may be
due to reworking of older sediments in the basin, up to 45 kyrs old, as suggested by the oldest obtained ages (Fig. 5).
Nevertheless, the relatively low Gelman and Rubin Reduction Factor (<1.05), the convergence test and the stable log-posterior



time-series suggest a relatively reliable age model and good fitting (Fig. 5A). According to this model, the sedimentation rate in the core 2013-04 comprises the last 26.4 cal kyr BP. For core 2012-PL1, the available ages correspond only to the deeper and central section of the 1.4 m-long core (0.73 to 1.39 m, Table 1) and the analytical errors of the ages are up to 50% in some cases (LFP-6 and LFP-7, excluded in the model), so ages must be taken with caution. The age of gypsum crystals in the sediment provided by Obert et al. (2022) ranges from 34±1.5 to 47±2.7 cal kyrs BP, with no age reversals considering the large analytical errors of some ages. The chronological model for core 2012-PL1 suggests that the deepest cores section is at least 40-50 kyr old (Fig. 5B).

**Table 1. Chronological data from FdP playa-lake cores 2013-04 and 2012-PL1 (Höbig et al., 2016 and Obert et al., 2022, respectively, see locations in Fig. 3). Radiocarbon ages from Höbig et al. (2016) were calibrated by OxCal (Bronk Ramsey, 2009) and IntCal 13 curve (Reimer et al., 2013). In core 2012- PL1, "dc" refers to the activity ratios corrected with the conventional approach (see Obert et al., 2022).**

| Core | Location | Sample ID | Depth (m) | Material | Conv. age (a BP) | Cal. age (a BP) | 2σ(cal a BP) |
|------|----------|-----------|-----------|----------|------------------|-----------------|--------------|
| 2013-04 | SW lakeshore | Beta-366926 | 7.71 - 7.72 | bulk sediment | 24,810 ± 150 | 29,790 | 30,160 to 29,420 |
| 2013-04 | SW lakeshore | Beta-365743 | 8.38 - 8.39 | bulk sediment | 38,170 ± 420 | 42,625 | 43,180 to 42,070 |
| 2013-04 | SW lakeshore | Beta-365744 | 9.93 - 9.94 | bulk sediment | 39,690 ± 500 | 43,735 | 44,500 to 42,970 |
| 2013-04 | SW lakeshore | Beta-365745 | 10.72 - 10.73 | bulk sediment | 32,450 ± 240 | 36,980 | 37,420 to 36,540 |
| 2013-04 | SW lakeshore | Beta-365746 | 11.45 - 11.46 | bulk sediment | 31,450 ± 220 | 35,870 | 36,510 to 35,230 |
| 2013-04 | SW lakeshore | Beta-365747 | 13.69 - 13.70 | bulk sediment | 22,290 ± 100 | 26,800 | 26,970 to 26,630 |
| 2013-04 | SW lakeshore | Beta-386843 | 9.91 - 9.93 | oogonia + seeds | 37,750 ± 360 | 42,063 | 42,520 to 41,605 |
| 2013-04 | SW lakeshore | Beta-386844 | 10.99 - 11.00 | charcoal | 3570 ± 30 | 3865 | 3835 to 3895 |
| 2013-04 | SW lakeshore | COL2742.1.1 | 1.82 - 1.83 | plant/wood | 212 ± 53 1 | 763 | 1884 to 1642 |
| 2013-04 | SW lakeshore | COL2740.0.1 | 3.19 - 3.20 | pollen | 29,053 ± 154 | 31,598 | 31,919 to 31,277 |
| 2013-04 | SW lakeshore | COL2737.0.1 | 5.79 - 5.80 | pollen | 25,464 ± 345 | 28,499 | 29,016 to 27,982 |
| 2013-04 | SW lakeshore | COL2735.0.1 | 7.50 - 7.52 | pollen | 16,825 ± 163 | 18,092 | 18,415 to 17,769 |
| 2013-04 | SW lakeshore | COL2734.0.1 | 8.38 - 8.40 | pollen | 43,510 ± 515 | 45,122 | 46,719 to 43,525 |
| 2013-04 | SW lakeshore | COL2733.0.1 | 9.20 - 9.22 | pollen | 15,008 ± 226 | 16,290 | 16,573 to 16,007 |
| 2013-04 | SW lakeshore | COL2731.0.1 | 9.93 - 9.96 | pollen | 31,999 ± 190 | 34,113 | 34,472 to 33,754 |
| 2013-04 | SW lakeshore | COL2730.0.1 | 10.12 - 10.14 | pollen | 12,491 ± 123 | 12,822 | 13,187 to 12,457 |
| 2013-04 | SW lakeshore | COL2729.0.1 | 11.44 - 11.45 | pollen | 19,159 ± 173 | 21,007 | 21,317 to 20,697 |

| Core | Location | Sample ID | Depth (m) | Material | $^{230}$Th/U ages | $^{230}$Th/U ages$^{dc}$ |
|------|----------|-----------|-----------|----------|-------------------|--------------------------|
| 2012-PL1 | Lake centre | LFP-1 | 0.73 | gypsum aggregates | 40,400 ± 1,500 | 34,000 ± 1,500 |
| 2012-PL1 | Lake centre | LFP-2 | 0.825 | gypsum aggregates | 35,700 ± 6,700 | 29,000 ± 6,200 |
| 2012-PL1 | Lake centre | LFP-3 | 0.875 | gypsum aggregates | 49,700 ± 2,300 | 40,700 ± 2,200 |
| 2012-PL1 | Lake centre | LFP-4a | 0.975 | gypsum aggregates | 70,500 ± 5,600 | 58,300 ± 5,300 |
| 2012-PL1 | Lake centre | LFP-4b | 0.975 | gypsum aggregates | 58,700 ± 1,600 | 49,300 ± 1,700 |
| 2012-PL1 | Lake centre | LFP-5 | 1.075 | gypsum aggregates | 66,200 ± 1,700 | 47,700 ± 2,700 |
| 2012-PL1 | Lake centre | LFP-6 | 1.325 | gypsum aggregates | 113,600 ± 5,900 | 76,000 ± 30,000 |
| 2012-PL1 | Lake centre | LFP-7 | 1.395 | gypsum aggregates | 113,700 ± 4,000 | 78,000 ± 30,000 |

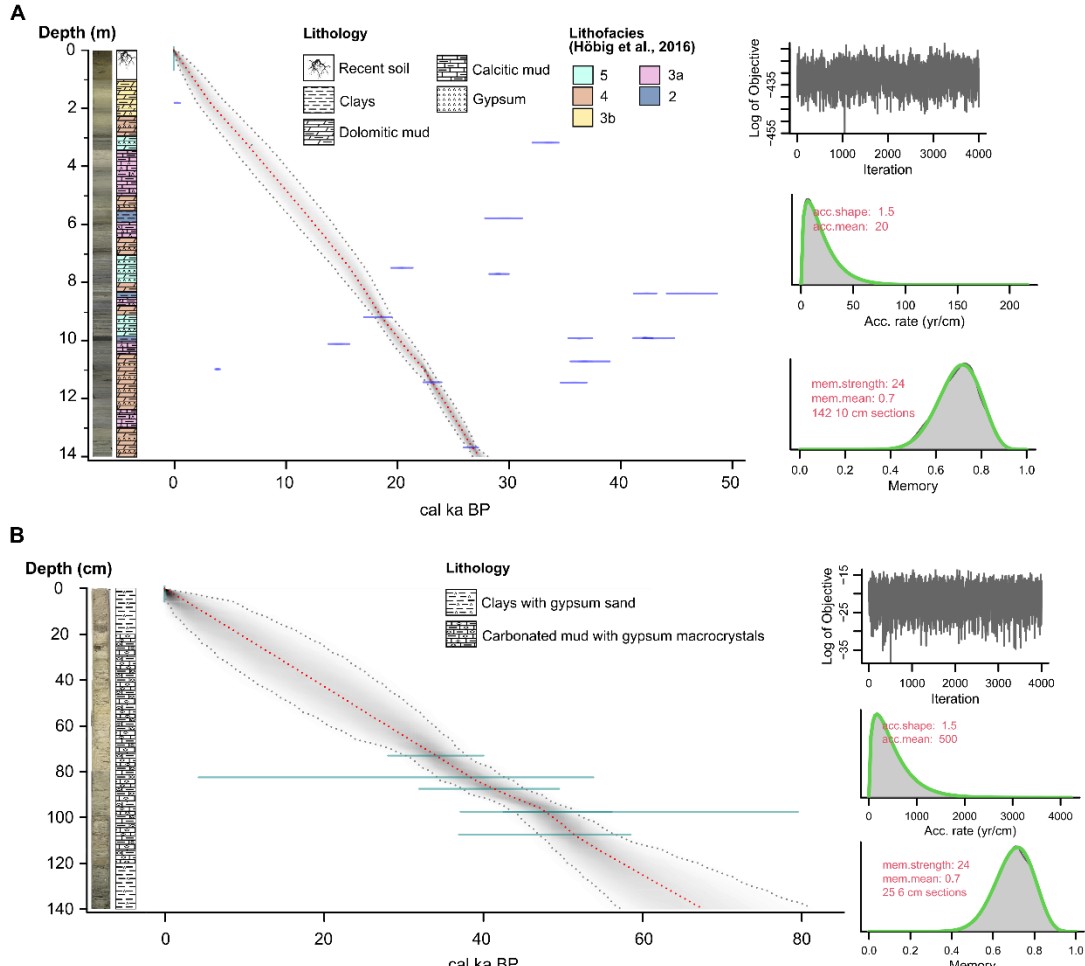

**Figure 5: (a) Sedimentary sequence, lithology, lithofacies interpreted by Höbig et al. (2016) and age-depth model with a 95% confidence range (grey shade) and single best model (red median line) of core 2013-04. The calibrated ¹⁴C dates are presented together with their uncertainty in blue. The iterations plot shows a rather stationary distribution. The prior and posterior distributions of the accumulation rate are depicted on the accumulation rate plot as green and grey lines, respectively. The previous and posterior model memory distributions are shown as green and grey lines, respectively, in the memory plot. (b) Sedimentary sequence, lithology and age-depth model (95% confidence interval and red median line) of core 2012-PL1. For age model references, see Table1.**

The sediments in core 2013-04 (Fig. 6A) are essentially composed of fine-grained materials, primarily clays and carbonate (calcitic, aragonitic or dolomitic) muds, along with evaporites represented by gypsum and halite. Gypsum shows several crystal morphologies, including single or twinned micro- and macrolenticular crystals (which often developed interstitially in the clay or mud), mm to cm-scale prismatic crystals and subangular to rounded gypsum sand grains, interpreted previously by Höbig et al. (2016) as "detrital" gypsum (Fig. 6). These authors also assessed different lithofacies in core 2013-04 based on macro- and microscopic observation and geochemical data (Fig. 6A). Lithofacies 2 contains layers of cm-thick massive clays with microcrystals of lenticular gypsum, while lithofacies 3a and 3b are mainly formed by beds of carbonate mud with



microlenticular gypsum crystals that sometimes are embedded into the mud. In turn, centimetre-thick layers of dolomitic mud and gypsum sand grains, both with interstitial mm-size lenticular gypsum crystals, are the principal components of lithofacies 4 and 5, only differentiated by the abundance of gypsum sand layers, more frequent in lithofacies 5. Less abundant are prismatic

crystals and cm-size macrocrystals of lenticular gypsum.

In core 2012-PL1, the upper 20 cm are dominated by laminated grey clays intercalated with mm-thick layers of gypsum sand. Between 20 and 120 cm depth, the core is composed of pale carbonate mud with embedded prismatic gypsum macrocrystals of centimetre-scale (Fig. 6). The deepest section shows a similar composition to the top, but the clays have a darker grey colour.

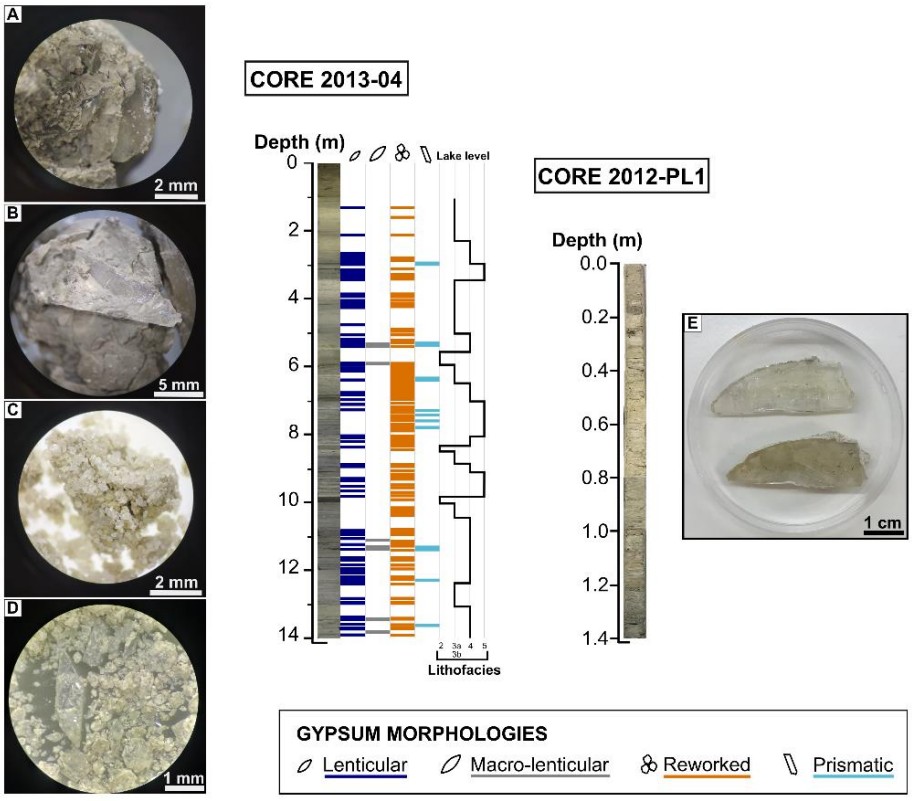


**Figure 6: Distribution of gypsum morphologies in cores 2013-04 and 2012-PL1. A. Lenticular crystals. B. Macro-lenticular crystals. C. Rounded sand-size gypsum grains (reworked). D. Prismatic crystals. E. Prismatic macro-crystals. The FdP playa-lake level reconstruction by Höbig et al. (2016) based on sedimentary facies is shown for comparison.**

## 4.3 FdP hydrology

### 4.3.1 Pleistocene-Holocene water level evolution of the FdP playa-lake

The results of the hydrological modelling of the FdP playa-lake over the past 35 kyrs are shown in Fig. 7. This includes modelling using WHC of 75 and 50 mm. For these calculations, first we considered that both the FdP watershed (W) and



average flooded surface (AFS) remained constant. When using WHC of 75 mm, the modelled runoff for the FdP watershed during the entire period is relatively low (Fig. 7A). Consequently, the lake was probably dry during most of the time and

increases on the water level (ΔV) did not reach 0.5 m except for a humid period from 32 to 30 kyrs BP (Fig. 7A). We considered that this model is not realistic because with similar precipitation and evapotranspiration values (350-700 mm for P and close to 1000 mm for ETP) in the historic period from 1983 to 2022, much higher runoff values were observed (Rodríguez-Rodríguez et al., 2016; Fig. 7). Consequently, WHC of 75 mm was discarded for additional analyses. When using a WHC of 50 mm, runoff values in the FdP watershed during some periods are like present values (ca. 10-15% of precipitation). However, the

increase on the lake level (ΔV) is always below 2 m (Fig. 7B). Maximum values were obtained between 32 and 30 kyrs, when the ΔV was up to 1.8 m, and from 10 to 4 kyrs, with values ranging from 0.5 to 1.8 m (Fig. 6B). ΔV during the last 1,000 yrs ranged from 0.5 to 1 m, which is equal to the current ΔV FdP (Figs. 7 and 8).

The FdP playa-lake probably behaved as a seasonal lake, then ΔV would be equal to the maximum water level. However, during rainy periods, it probably behaved as a permanent lake. In this case, to obtain the maximum water lake level, we

recalculated the AFS for every year during extremely rainy periods of 50 yrs (P =700 mm, ETP = 850 mm and runoff = 75 mm; Fig. 7B), depending on the water level reached in the previous year (Fig. 9). We obtained that the FdP water level never surpassed a height of 5 m over the lakebed. Because of the flat bathymetry of the FdP playa-lake, the W/AFS relationship used to calculate the BD decreases from 11.1 to ca. 3 (i.e., W area tripled in size the AFS area) when the FdP water level increases from 1 to 5 m (Fig. 10). Consequently, the water level stabilizes at ca. 5 m, when water inputs are equal to water outputs (Fig.

9). Given that there is not a clear maximum water level, Figure 10 shows the FdP flooded area from 1 to 6 m above the lakebed. The contour of the lake deposit outcrops is also indicated.

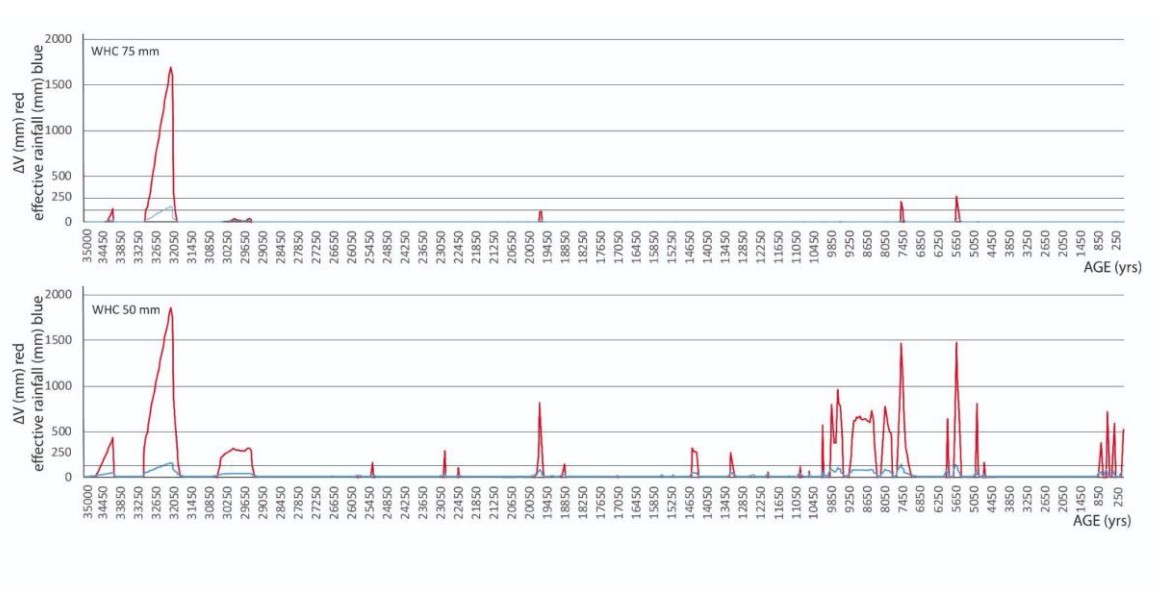





**Figure 7: Model results showing ΔV in the FdP playa-lake (in red) and effective rainfall (in blue) using WHC of 75 mm (a) and 50 mm (b).**

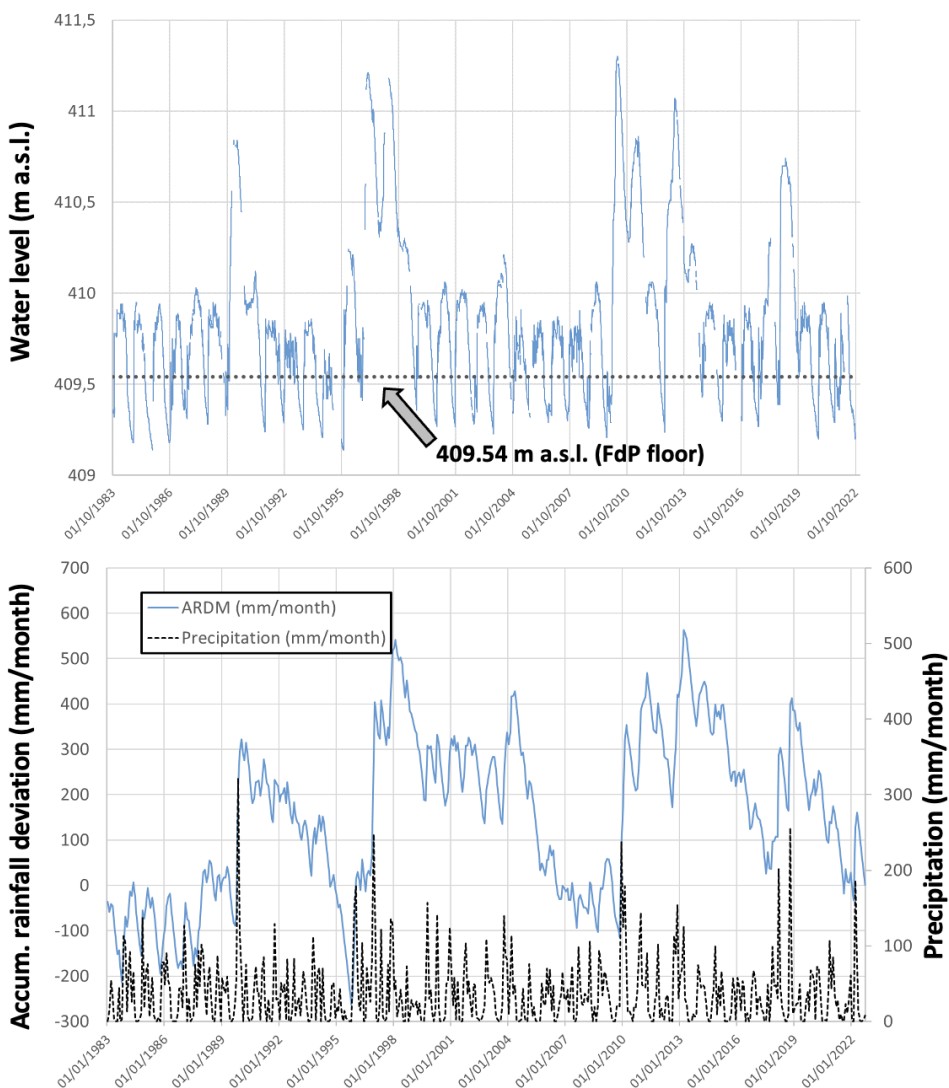

**Figure 8: Variations on the FdP playa-lake water level from 1983 to 2022. These 4 decades include both dry and wet periods.**



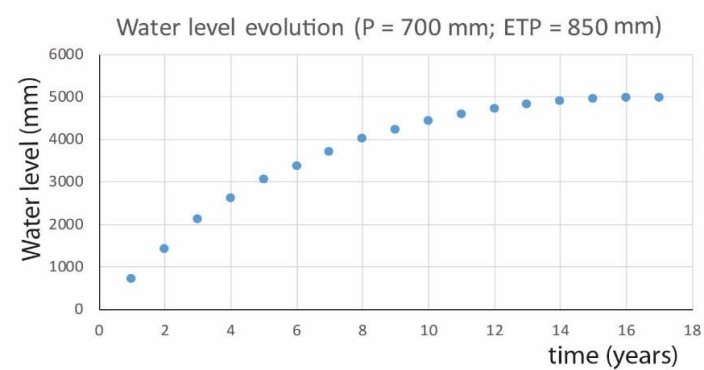

**Figure 9: Results of the FdP water level assuming stable humid conditions during 17 years, when the water level stabilizes at ca. 5 m above the lakebed.**

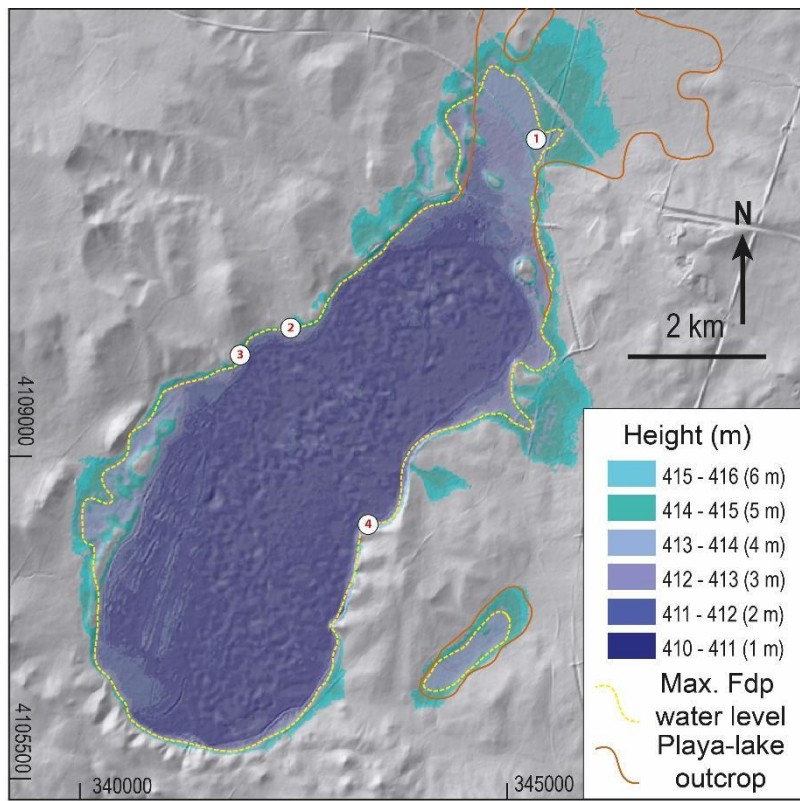

**Figure 10: Simulation of the FdP playa-lake flooded area (AFS) if the water level reaches 1 m to 6 m using the current topography. The boundary of the Pleistocene to Holocene FdP playa-lake sediments is also displayed (See section 3.2. for methodology). The location of piezometers 1 to 4 is shown.**

Several piezometers are installed at different depths to investigate the behaviour of the hypersaline groundwater brine below the playa-lake floor (Fig. 11). The distribution of electrical conductivity values of water in the studied profiles remained





relatively constant during the study period (1993-1998) in the different piezometers. The measurements in P2 and P3 reveal a transition zone approximately 1 - 2 m thick between the water from local rainwater and runoff from the catchment, which remains in the upper level because of its lower density, and the brine beneath FdP (Fig. 11). This transition zone is a result of mechanical dispersion and molecular diffusion processes. Within the transition zone, the theoretical Ghyben-Herzberg interface is located.

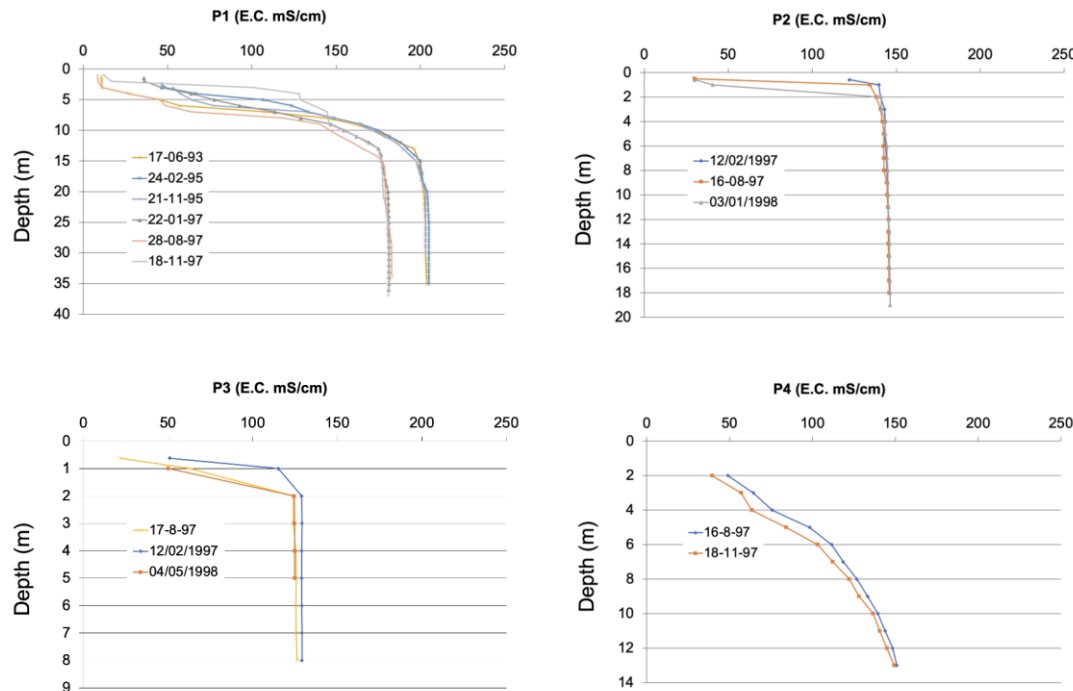


**Figure 11: E.C. profiles of the groundwater in the piezometers shown in Fig. 10. Note that P1 is in an area nowadays outside the current playa-lake basin. Differences in E.C. between 1993/95 and 1997 profiles are attributed to differences in precipitation. In P1, P2 and P3 a sharp freshwater – brine interfase can be detected, whereas in P4, the southernmost piezometer, the interfase is not so clear.**


## 5    Discussion

### 5.1 Factors controlling the FdP geometry and evolution

Here we propose a hydrological model for the FdP playa-lake based on a water balance simplified to precipitation and surface

and subsurface runoff from the watershed as water inputs and evapotranspiration (ET) as water outputs (Rodríguez-Rodríguez et al., 2016). According to this model, the watershed area is in equilibrium with the FdP average flooding surface (i.e., maintaining a constant W/AFS ratio) under almost unchanging climate conditions (precipitation and temperatures that control





the potential evapotranspiration). We considered that endorheic watershed size maintained along the FdP lifespan, along the last 35 kyrs. Some plausible increases/decrease on the FdP watershed (<10 km$^2$, 6% of the current watershed) would not imply

significant increases on the W/AFS relationship (Fig. 12).

Precipitation (P) and ET values have changed during the past 35 kyrs in southern Iberia: P ranged from 350 to 700 mm and ET varied between 650 to 1000 mm (Fig. 4). We chose the wettest period to calculate the highest expected water level (at approx. 6 kyrs before present: P = 700 mm and ET = 820 mm; Fig. 6) and we made iterative calculations by changing the W/AFS relationship during 50 yrs (Fig. 9). Our calculations suggest that the maximum feasible lake water level under such

climate conditions is 5 m. However, the FdP water level probably never reached 5 m above the lakebed because of the variability of the Mediterranean climate conditions (dry periods alternate with wet periods). More than 10 humid years are needed to obtain this value (Fig. 9). Moreover, the lake level during the past 30 yrs never exceeded 2 m (Fig. 8). Then, most playa-lake sediments should have been deposited in the lake shores between 0 and 5 m above the lakebed. Nevertheless, Pleistocene-Holocene FdP sediments are distributed from 0 m to more than 15 m above lakebed, NE-ward of the playa-lake

(Figs. 3A, 3B, 10 and 12). In this regard, previous works have interpreted that some sandstone outcrops located to the S of the playa-lake, are lacustrine terraces, which are currently at 10 m above lakebed due to climate changes (Höbig et al., 2016). However, this outcrops show features pointing to a high energy environment between the continent and the ocean: angular clasts of a wide variety of sources and a calcareous cement (see Pliocene conglomerates and sandstones in Fig. 3). These rocks are similar to other continental-marine transitional Pliocene deposits found within the ADZ (Cruz-Sanjulián, 1991; see stars

in Fig. 12). Consequently, we interpreted these sediments as the playa-lake basement instead of FdP lake sediments.

The presence of playa-lake sediments at more than 10 m above lakebed to the NE suggests a displacement of the FdP playa-lake towards the SW from the Late Pleistocene to the Holocene. Such behaviour is also consistent with the analyses of the EC profiles made in the piezometers (Fig. 11). The development and formation of a sharp transition zone between the underlying groundwater brine and the upper brackish-to-fresh discharge is detected in EC profile. As commented before, this chemocline

is associated with fluctuations in the water level of the playa-lake due to both seasonal drying and climatic cycles, as well as variations in the piezometric level of the aquifer. The observations indicate the existence of a brine reflux from the playa-lake floor beneath the transition zone, like what has been studied in well-documented analogous playa-lakes elsewhere (i.e., the Tyrrell Basin; Macumber. 1992) and also in this same system (Kohfahl et al., 2008).

The presence of a well-defined interface 1 to 5 m wide towards the north of the playa-lake (P1; see Fig. 11), in an area where

the FdP current playa-lake floor is absent, could indicate a potential migration of the lacustrine basin from the N-NE to the S-SW. This hypothesis is supported by two observations:

1.    The interface is highly stable over time; therefore, once formed, it would take a considerable amount of time for it to be washed away.

2.    In piezometer 4, located further south, the interface is considerably weaker, suggesting that the playa-lake floor has

not existed in this position for that long.





This finding supports the hypothesis of a lacustrine basin migration over time, potentially caused by tectonic factors. Indeed, the SW-ward movement of the FdP playa-lake is consistent with the kinematics of recent faults which determine its boundaries. (Fig. 12; Jiménez-Bonilla et al., 2023). Thus, the left-lateral component of the La Nava fault would have induced the SW-ward displacement of the main FdP depocenter. Additionally, the relative uplift of the NE edge of the FdP playa-lake is compatible

with its SW tilting, associated with the downthrowning of the Las Latas fault hanging wall. According to this, the normal dip-slip component of both La Nava and the Las Latas faults would have produced the FdP relative sinking of the FdP playa-lake, preventing its siltation and its capture from Mediterranean streams (e.g., Tinajas stream; Fig. 12). Therefore, the long lifespan of the FdP playa-lake (>30 kyrs), which is unusual for playa-lakes in S Spain (compare with, for example, the 11 kyrs-old Ballestera playa-lake; García-Alix et al., 2022), would have been favoured by the tectonic activity in this area. It is likely that

the relict La Nava playa-lake (Figs 3A, 3E and 12) was formerly part of the FdP playa-lake and was subsequently separated by uplift of the La Nava fault footwall. In summary, the FdP playa-lake geometry and evolution seem to be influenced, since the beginning, by the recent to active tectonic activity in the area, governed by the transpressive kinematics of the TSZ and the ABSZ. This tectonic scenery would be responsible for the NW-SE relief segmentation that has generated the ADA, favouring the inception of several endorheic basins there. Focusing on the FdP playa-lake, and according to our results (Fig. 5), its

inception occurred at least 50 kyrs ago, being its further evolution controlled by faults linked to the transpressive kinematics. In this regard, focal mechanisms of earthquakes and geomorphological studies support the current activity of both shear zones (Jiménez-Bonilla et al., 2015; 2023). Consequently, active tectonics seems to be a key factor in the shaping of these wetlands within this Betics fold-and-thrust belt segment.



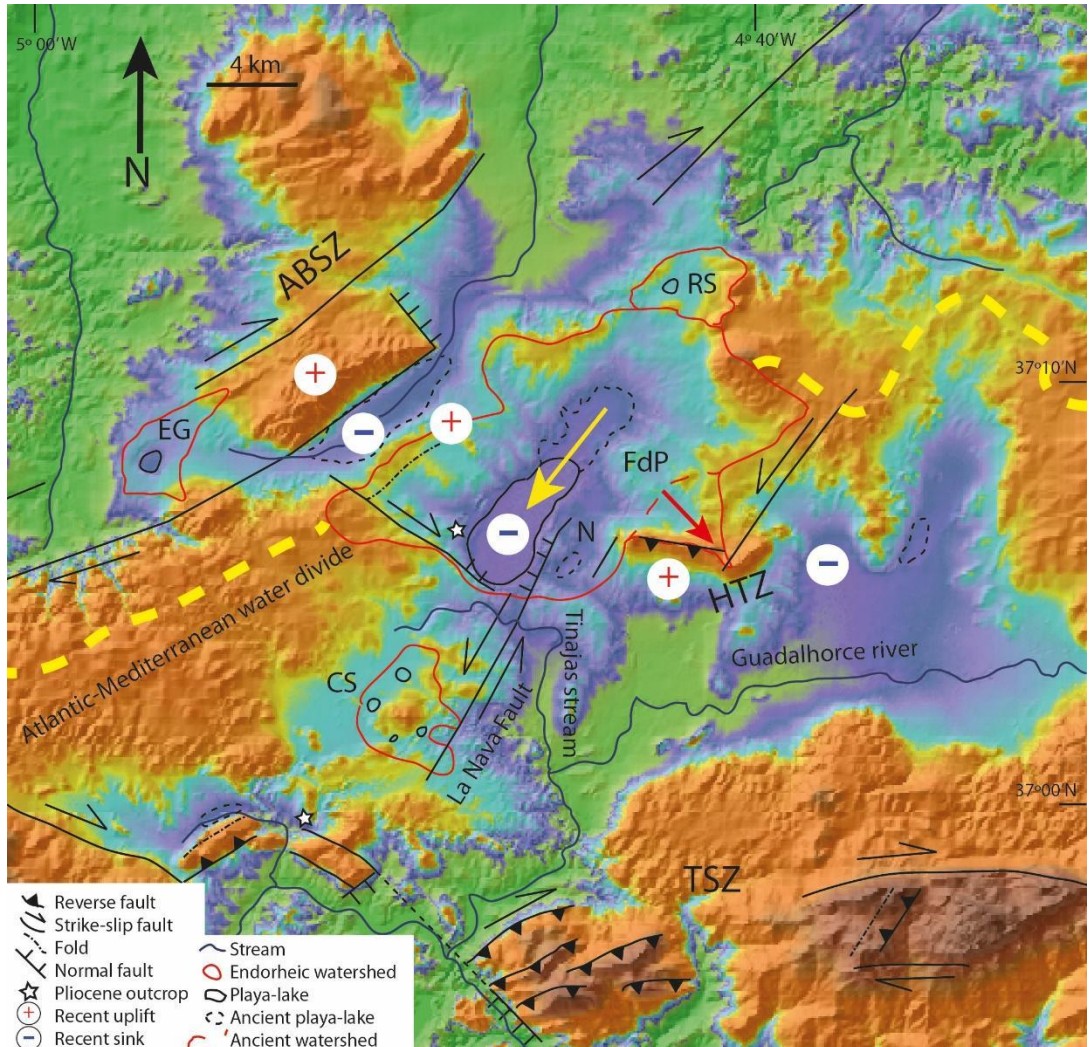

**Figure 12: DEM of the ADZ, where the Atlantic-Mediterranean divide (in yellow) is located in this sector of the Betics and the FdP watershed develops. Main structures are shown together with their estimated recent movements. EG: El Gosque, FdP: Fuente de Piedra, CS: Campillos system, RS: Ratosa system, N: La Nava relict playa-lake. Stars: location of Pliocene conglomerates and sandstones.**

### 5.2 The FdP evolution through its sedimentary record

Previous works based on radiocarbon dates suggested that the FdP playa-lake is the oldest one investigated in the area. Earliest ages are ca. 26 kyrs in the southern sector (Höbig et al., 2016) and may be earlier than ca. 50 kyrs in the central basin, as suggested by recent chronological analyses of gypsum (Obert et al., 2022). Moreover, drills did not reach the contact between the lacustrine sediments and the basement (Triassic rocks), so the FdP inception was probably even earlier.

In our work, the gypsum deposits observed in the core from the SW shore of the FdP playa-lake (2013-04; Figs. 3A and 4A) differ largely from those observed in the core taken from its centre (2021-PL1, Figs. 3A and 4B). The SW gypsum deposits



exhibit a wide range of crystal variations and sizes, with majority of fine grains, while only selenite crystals larger than 1 cm interbedded with mud are found in the lakebed in its central part (Fig. 5). The morphology of gypsum crystals in sediments of

continental environments can offer hints to understand their depositional setting (Cody and Cody, 1988). According to these authors, larger prismatic crystals growth is chiefly independent of temperature, and they develop in environments with relatively low concentrations of dissolved organic compounds. Also, larger gypsum crystals are expected to grow under relatively stable conditions. These stable conditions can be attributed to periods of water level high stands and a more permanent playa-lake level (e.g. from 10 to 7.5 kyrs ago; Fig. 3). Despite the uncertainty in the ages obtained by Obert et al.

(2022), and assuming that the results of their challenging U-Th measurements are correct, the age of the sediments in the central basin is older than 48 kyrs but may be as old as 98 kyrs, according to the age uncertainties. Importantly, sediments at about 0.7 m below the playa-lake floor in its central part are 34±1.5 kyrs (2012-PL1; Figs. 3A and 4B). This chronological model suggests that, either the sedimentation rate for the upper 0.7 m was considerably low (0.02 mm/yr) during the past 34 kyrs or, instead, part of the sediments that were deposited in the central area of the FdP playa-lake during the Holocene may

have been eroded/dissolved more recently. The topographic and tectonic uplift of the northern part of the basin and subsidence of its southern part may have favoured exposure of the northern sediments to erosion because of slow tilting of the basin to the south, which is congruent with the kinematics of its faulted boundaries (Fig. 12). Remobilization of older lacustrine sediments because of waves produced by wind during highstand periods, and/or by aeolian erosion during lowstand periods would have resulted in lakebed exposure and, consequently, migration of "reworked" gypsum grains to other parts of the basin. This agrees

with the current slope showed by the lacustrine deposits at the NE corner of the FdP playa-lake, which reaches 1.5% (Fig. 3B). Höbig et al. (2016) attributed the presence of rounded sand-size gypsum grains found throughout the entire length of core 2013-04 from the SW shore to periods when playa-lake water agitation caused sediment reworking and transportation from shallower to deeper areas. This is consistent with the hypothesis of lacustrine basin migration towards the SW as the elevation of the northern area could have produced a more energetic environment in the SW (Fig. 12). In this scenario, sediments from

the NE may be continuously eroded, transported and deposited towards the SW, at different distances from the source area. Unfortunately, no scientific drilling reaching the contact between the lacustrine sediments and the underlying Triassic material has been conducted in FdP to date, either in the margins or in the central part of the basin. Further investigations on the FdP sedimentary sequence may shed light on the geometry of this contact. According to our hypothesis, this contact may be at a deeper position in the southern playa-lake sector than in its northern part, because of the southward tilting of the basin.


## 6    Conclusions

Previous studies attributed the presence of lacustrine deposits surrounding the Fuente de Piedra playa-lake to climate

fluctuation (Höbig et al., 2016). In this work, we applied a hydrological modelling for the Fuente de Piedra (FdP) playa-lake



during its lifespan (> 35 kyrs). To this end, we reconstructed direct evaporation, ET and runoff from P and temperature previously published and we calculated the increase on the water level from the playa-lake bottom every 50 years. The increase on the water level never reached 2 m and iterative calculations show that the maximum water level is lower than 5 m. However, we detected lacustrine deposits more than 10 m above the lakebed, then additional forces apart from climate changes, are
evaluated here. According to geological results, we propose the SW-ward displacement of the FdP playa-lake during the Pleistocene-Holocene because of recent tectonic activity. This displacement is favoured by the activity of a left-lateral dominated La Nava fault and the dip-slip dominated Las Latas fault, that contribute to the downthrown and tilting of the block where the FdP playa-lake is located. Other lines of evidence support this hypothesis:

1. There are lacustrine sediments related to the current FdP playa-lake more than 15 m above the lakebed at the NE
corner of the playa-lake. In the E of the FdP playa-lake an isolated lacustrine deposit is located 3 m above the present FdP lakebed. The uplift and isolation of the patch is compatible with the La Nava fault kinematics.

2. The presence of a well-defined water interface towards the north of the FdP basin, above the maximum FdP water level, also indicates a SW migration of the FdP depocenter.

3. The morphologies of gypsum found in the sediments indicate the existence of primary gypsum formed in-situ during
certain periods. Significant amounts of reworked gypsum, probably transported from northern parts of the basin, were also deposited throughout most of the time covered by this sedimentary sequence, especially to the S. The SW-ward tilting of the playa-lake may have facilitated this process.

Our study confirms the importance of multidisciplinary investigations to understand the inception and development of wetlands and saline playa-lakes in subtropical arid conditions.

## Data availability
All raw data can be provided by the corresponding authors upon request.

## Author contributions
AJB, LM, MRR, FG and SM planned the campaign; SM, MRR, AJB and LM performed the measurements; AJB, MRR,
LM, FG, IER, MD, KR analyzed the data; AJB, FG, LM and MRR wrote the manuscript draft; IER, MDA and KR reviewed and edited the manuscript.

## Competing interests
The authors declare that they have no conflict of interest.

## Acknowledgements



We thank A. Lupión (Director of the Patronage of the Natural Reserve of Fuente de Piedra playa-lake) for support during field surveys and for providing historical monitoring data of the lake.

**Funding**

This study was supported by the (1) Tectonic conditioning and climate change effects on the hydrogeological evolution of
wetlands and playa-lakes in the southern Spain research project (University of Pablo de Olavide), (2) GYPCLIMATE research project (PID2021-123980OA-I00) Spanish Ministry of Economy and Competitiveness - Regional Development European Fund (FEDER), (3) the project PGC2018-100914-B-I00, funded by the Ministerio de Ciencia e Innovacíon (Spanish Government)/AEI/10.13039/501100011033/ERDF, (4) the project UPO-1259543, funded by the Consejería de Economía, Conocimiento, Compañías y Universidad (Andalusian Government)/ERDF and (5) "Monitorizacion hidrológica y
modelización de la relación laguna-acuífero en los mantos eólicos de Doñana. Seguimiento y ampliación del inventario" (Agreement between the Guadalquivir River Basin Authority and the University Pablo de Olavide). Dr. F.G acknowledges the Ramón y Cajal fellowship, RYC2020-029811-I and the grant PPIT-UAL, Junta de Andalucía-FEDER 2022-2026 (RyC-PPI2021-01). L.M. was funded by the FPU21/06924 grant of the Ministerio de Educación y Formación Profesional of Spain.

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
