# Peer review of "Late-Quaternary hydrological evolution of Fuente de Piedra playalake (southern Iberia) controlled by neotectonics and climate changes"

_Hydrology and Earth System Sciences, 2024_

## Author Response (AR1)

29 September, 2024

Dear Editor:

Please convey our gratitude to the reviewers for their detailed reports, which will considerably improve the manuscript. In the new version, you will find that all issues were tackled.

**Regarding comments from Reviewer #1**

**Specific comments**

> The title implies that climate changes are a controlling factor of the playa-lake system. However, the discussion mostly focusses on the influence of neotectonics. There is some discussion of the (maximum) lake water level in regards to the high variability of the Mediterranean climate conditions (Line 386) and short statement on the existence of a more permanent playa-lake level from 10 to 7.5 kyrs ago (Line 448f). The identification of this previously unknown parameter in the formation/preservation of the playa-lake is an important finding, but it also allows for a reevaluation of the influence of the climatic conditions. Therefore, in the discussion of the evolution of the lake the authors should elaborate the role of climate changes or climate variability for the evolution of the lake.

*Answer*: **We appreciate the reviewer´s comment and agree that the discussion is mostly focused on the neotectonics. In the new version of the MS, we include a more detailed discussion about climatic and hydrological forces. We better discuss the role of the climate variability on the evolution the playa-lake hydroperiod (e.g.: "During some extremely wet periods described by Camuera et al. (2022) such as WMHP-3 (39–29 kyr BP), WMHP-2 (27–18.5 kyr BP; WMHP-2.2 at 27–25 kyr BP and WMHP-2.1 at 23–18.5 kyr) and WMHP-1 (15.5–5 kyr BP), the FdP playa-lake probably enlarged its hydroperiod by over 80%. Indeed, it could have behaved as a permanent lake during short periods, like other lakes in this area at present (e.g. Amarga Lake; Jiménez-Bonilla et al., 2023)." in the discussion section. Additionally, we have included more climatic and hydrological information. However, as suggested by the reviewers, because of the importance of the neotectonics in our study, we changed the title to "The role of neotectonics and climate variability on the Holocene hydrological evolution of the Fuente de Piedra playa-lake (southern Iberia)".**

> The integration of the lake evolution into the regional hydrogeologic model could be emphasized more in the abstract.

*Answer*: **We agree with the reviewer and have added a sentence at the end of the abstract to summarize our conclusions: "Consequently, the flooded surface of the FdP remained largely constant and in equilibrium with climate variables and its**

**watershed throughout its lifespan. The SW-ward displacement of the flooded surface was caused by recent tectonic activity".**

Some figures need slight modifications (see technical comments below)
*Answer*: **We modified the figures according to the reviewer´s comments (see answers below)**

Sections 5.1 and 5.2 in the discussion chapter should switch places. In my opinion, the local playa-lake evolution should be placed before the integration of the findings into a regional model. When the chapters are switched, some minor internal adaptations of the individual text blocks might be needed.
*Answer*: **We agree with the reviewer and have modified the order of paragraphs in sections 5.1 and 5.2.  We have switched the place of the sections, and we made small changes within them.**

**Technical corrections:**

The variables in the text could be formatted in italics to increase the readability.
*Answer*: **Thanks, we have made modified the format of the variables in the current MS.**

Line 32: Is it led or lead?
*Answer*: **We mean "lead" and have been changed it in the new MS version.**

Line 96: Rivers are not shown in Figures 1 and 2
*Answer*: **Thanks, we have included the drainage network in Figs. 1 and 2 in the new MS version, along with the names of the main rivers.**

Line 99: The numbers and letters in brackets make it hard to read. Maybe you don't need the numbers.
*Answer*: **As suggested by the reviewer, we have deleted number levels in Figs. 1 and 2.**

Line 107ff: If I understand correctly, this sentence refers to the sediments that cover the Subbetics. It is not entirely clear that you refer to with "They" at the beginning of the sentence. This sentence appears a bit out of context in this position, I would rather place it somewhere after Line 114 in context with the description of the other sediments.
*Answer*: **In order to make clear that we refer to the Subbetic units here, we have changed "They" by "Subbetic units".**

Line 114: Maybe add "Upper Miocene shallow-marine deposits testify [...]" to make it clear that these sediments are different from the previously described Subbetics.

*Answer*: **As suggested, this clarification has been incorporated in the current MS version.**

Line 116: "[...] that are currently deformed and [...]". Does this refer to active tectonics?

*Answer*: **Active tectonics comprises geological structures that were active during the Holocene or, at least, during the Quaternary. In this case, we mean that previous regional maps show that some geological structures affect upper Miocene rocks. It does not mean current active tectonics, but these structures were active after Tortonian. To make it clearer, we have changed the sentence to: "Previous geological maps show that these sediments are deformed by open folds and faults and often lie unconformably over Subbetic units (Flinch and Soto, 2017;2022)".**

Line 231: erase one "area."

*Answer*: **Changed in the new version of the MS version.**

Line 395ff: The switching between playa-lake basement, FdP lake sediments and playa-lake sediments or FdP playa-lake is a bit confusing at this section. It would be better to stick with one fixed term e.g. FdP playa-lake sediments for the whole manuscript.

*Answer*: **The referee is right, we have aimed to clarify and simplify the concepts. In the current MS version, we consistently refer to the water body as the "FdP playa-lake" and to its sediments as "FdP playa-lake sediments".**

Figure 2: The extent of the ADZ and the letters of the playa-lakes are hard to see. Please make them a bit darker.

*Answer*: **Changed it accordingly.**

Figure 3: This figure is nice and fully packed with information. However, I have some suggestions for this figure as I had some difficulties differentiating the lithologies and finding the traces of the cross sections:

The traces of the cross-sections are difficult to spot, make them thicker and the lettering a bit larger.

*Answer*: **Changed it accordingly.**

The legend is quite small and there is no entry for the borehole locations.

*Answer*: **We enlarged the legend font and we included the borehole locations.**

Maybe because of the small sized legend, but also due to the tight color palette, the lithological units are hard to tell apart. I had difficulties with the rose colored lithologies (e.g. evaporites from dolostones, sandstones and alluvial sediments), but also the green colored marly-limestones and erosive pediments could use a better separation (the latter could be bordered by a thicker line to represent the unconformity) and also the blue colored limestones, alluvial fans and lunettes.

*Answer*: **We have changed colors to make this figure more readable.**

Is it correct that the fold axis is represented by both, a dashed and a solid line in the erosive pediment? In line 269 you write that the fold is locally truncated by the erosive pediment, so do I understand correctly that in some places the erosive pediment is folded (solid line), in other places the hinge zone is truncated (dashed line)?

*Answer*: **Yes, that is correct. We use solid lines to indicate the erosive pediment seems to be folded, whilst we used the dashed line to mark the fold is truncated by the erosive pediment.**

One coordinate grid number on the left border is misplaced.
*Answer*: **Modified.**

Inset F is quite small please make it larger. As it is now it is hard to tell what is shown.
*Answer*: **We have enlarged it.**

Figure 4 and 7: On the x-axis I would put time (yrs BP) instead of age.

*Answer*: **We changed it accordingly.**

Figure 6: It is not entirely clear to me how to read the lake level scale in this figure. Is it the translation of the lithofacies (they are hardly readable) into relative water depth? Please indicate where high or low lake-level would be.

*Answer*: **We have made changes on this figure in order to make show the changes in the lake levels inferred from lithofacies.**

Figure 8: There is a different scale on the x-axis. Please make them the same size.

*Answer*: **We changed the x-axis to homogenize the scale.**

Figure 11: The last sentence in the caption would be better placed in the text.

*Answer*: **Now we have places this in the main text.**

**Regarding comments from Reviewer #2**

**We thank very much to Blas for these valuable comments. They allowed us to significantly improve the previous version of the MS.**

The figures are informative and well-designed, but I would suggest a few edits to some of the figures:

The size and colors of some lettering in Figures 2 and 3 could be changed to make them easier to read.

*Answer*: **Thanks. We have made some changes in both figures in order to make them more readable (see our answers to the comments from reviewer #1).**

Figure 1. It could include also a figure with main climate features of the region.

*Answer*: **We have added the main climates features in this figure.**

Figure 3. It could also include the topography, so the drainage and the delineation of the subbasins would be easier to visualize

*Answer*: **Figure 3 already includes a hillshade and some topographic points. We have added more topographic landmarks, including river names to make it more informative.**

Figure 4. It could include all paleoclimate periods, plus de main humid periods identified in El Padul record (Camuera et al. 2022). As an inset it could also include the correlation between recent SST in Alborán and weather stations close to FdP.

**See attached for the new Fig. 4**

*Answer*: **As the referee indicates, there is a lack of climatic information in the MS. We agree with that and have included other climatic periods in this figure in addition to the LGM. Regarding the correlation between recent SST in Alborán and weather stations close to the FdP: We made a correlation between the SST (using Cabo de Gata buoy) and land temperature (Antequera weather station) during an instrumental period of more than 20 years (2000-2023) and we obtained a correlation coefficient of 0.76. We added it in the new version of the MS.**

**See attached for Fig. A (for revision). Temperature in the weather station of Antequera (blue line) and in the Cabo de Gata buoy from 2000 to 2023.**

Figure 6. Any lake level inferences for central core based on gypsum crystals morphologies?

*Answer*: **Thanks. The lithology of core 2012-PL1 from the central FdP is not as well defined as in core 2013-04, from the lake shore. As stated in section 4.2, the upper 20 cm of core 2012-PL1 are dominated by laminated grey clays intercalated with mm-thick layers of gypsum sand. Between 20 and 120 cm depth, the core is composed of pale carbonate mud with embedded prismatic gypsum macrocrystals of cm-scale. Since we suspect that part of the lake sediments in the central/northern are of FdP has been eroded away and transported to the southward areas (see discussion), lake level reconstruction from gypsum morphologies in this core is not possible.**

Figure 7. To better understand the similarities with previous work it should also include the lake level reconstruction by Höbig et al (2016) and the paleoprecipitation and main humid periods by Camuera et al (2022).

*Answer*: **We included in figure 4 the lake level reconstruction by Höbig et al., 2016 to compare with the main humid periods described by Camuera et al., 2022.**

Figure 10. Include the location of the 15 m (NE) and 10 m (S) deposits.

 *Answer*: **Thanks, we have included both locations in the new MS.**

A final figure showing the depositional evolution of the lake, the paleoprecipitation, lake level reconstructions (both model and sediment cores), regional paleoclimate and regional and local tectonics would be a good summary of the manuscript.

  *Answer*: **Both reviewers suggested the addition of a final summarizing figure taking together tectonics, sedimentation, climatologic and hydrologic features. In order to accomplish such suggestions,  the new version of the MS, we made a simplified figure showing the FdP playa-lake evolution at different stages from 35,000 years until present.**

**See attached for Fig. 13: Summarizing sketch of FdP playa-lake evolution according to our tectonic, hydrogeological and sedimentological results.**

I have several suggestions regarding the integration of previous data and the interpretation of some results (see below).

**Paleoclimate.** Paleoprecipitation reconstruction for Padul pollen record has to be considered with caution, as the authors stated in the original paper. Camuera et al (2022) paleoprecipitation reconstruction showed several humid periods during the last 200 ka, some of them during the life span of FdP: WMHP-3 (39–29 kyr BP), WMHP-2 (27–18.5 kyr BP; WMHP-2.2 at 27–25 kyr BP and WMHP-2.1 at 23–18.5 kyr) and WMHP-1 (15.5–5 kyr BP). These periods should be discussed in the manuscript and marked in Figure 4. AS I mentioned before, in this Figure it would be helpful to indicate also all the paleoclimate periods, not only the LGM.

*Answer*: **Thanks. We have included some climatic periods in Fig. 4, according to the reviewer´s comment. In the discussion section, we have better explained the importance of climatic variability on the hydroperiod of the FdP playa-lake. We included this: "During some extremely wet periods described by Camuera et al. (2022) such as WMHP-3 (39–29 kyr BP), WMHP-2 (27–18.5 kyr BP; WMHP-2.2 at 27–25 kyr BP and WMHP-2.1 at 23–18.5 kyr) and WMHP-1 (15.5–5 kyr BP), the FdP playa-lake probably enlarged its hydroperiod to more than 80%. Indeed, it could have behaved as a permanent lake during short periods, as it currently occurs with other lakes in this area (e.g. Amarga Lake; Jiménez-Bonilla et al., 2023)."**

**Line 454 "As can be deduced from Fig. 8, FdP playa-lake has been dry almost every summer, during the period with instrumental records (1983 – 2022) except for two humid periods (1996-1997 and 2010-2011) in which the playa-lake remained flooded for two consecutive years. From the analyses of the data record obtained in the deepest part of the playa-lake's floor, during the daily record of the lake's hydroperiod, 9710 out of 14610 days, water level was higher than 409,54 m A.S.L. which is the altitude in which level drops to zero m (consequently, the playa-lake dry-out completely). So the average flooded period (hydroperiod) for this playa-lake, to date, is 66,5% (see Fig. 8 for details)."**

The SST reconstructed from the Alborán Sea site is considered comparable to FdP air temperature (Line 320). To increase the strength of the argument, the authors could explore the correlation between SST and land temperature (weather stations) during the instrumental record.

*Answer*: **As suggested by the reviewer, we have made a correlation between the SST (using Cabo de Gata buoy) and land temperature (Antequera weather station) during an instrumental period of 20 years (2000-2020) and we obtained a correlation coefficient of 0.76. We added it in the new version of the MS.**

**Hydrogeology.** The hydrological modeling requires "constant watershed surface area for past 35 kyrs" (Line 188). However, the authors invoke a clear role of neotectonics,

changing the basin configuration, as the depocenter shifted and , some areas - as La Nava subbasin - were also part of the watershed. These changes add some uncertainty to the model that could require further discussion.

*Answer*: **In the MS we explain that the FdP watershed mostly remained constant during its evolution (See: "Some plausible increases/decrease on the FdP watershed (<10 km$^2$, 6% of the current watershed) would not imply significant increases on the W/AFS relationship (Fig. 12)." Neotectonics partially affected the FdP watershed configuration but did not change its area, which mostly remained constant during the 35 Kyrs. Active faults mostly affected the FdP playa-lake (its flooded area).  As reviewer #1 suggested, we have clarified the meaning of FdP watershed and FdP playa-lake and we tried solving this confusion in the current MS version. Moreover, in order to make it clearer, now we say: "Although the watershed geometry could have changed, the FdP watershed area probably maintained."**

For the non - hydrogeology audience some concepts as "water holding capacity" could be explained a little bit more in detail beyond including a reference (Rodriguez Hernández et al., 2007).

*Answer*: **We have clarified this concept. We state now that: "The WHC is the amount of water a soil can hold without generating runoff and it depends on the soil texture".**

Although there are many references to previous work, there is no detailed hydrogeological synthesis from FdP. Both surface and subsurface watersheds are considered comparable (Kohfahl et al., 2008) but there is no description of the nature and extension of the aquifers and subsurface flows. The authors suggest possible flows from the carbonate aquifer of Mollina mountain range) Santillán stream; Fig. 3) and, most probably, groundwater inputs also from the Sierra de Humilladero mountain (line 143), but some information could be included about the volume and hydrologic properties of such aquifers and hydrogeological data ruling out long distance groundwater inflows.

*Answer*: **For our calculations, the runoff coming the karstic aquifers is included within the BD. We have included: "*BD* includes the groundwater coming from the karstic aquifers and the surface runoff coming from all the FdP playa-lake." to make it clearer.**

**"...although no springs have been observed in this carbonate aquifer. Both aquifers are intensively exploited at present (e.g., Rodríguez-Rodríguez et al., 2015), but in natural regime there are estimations of the recharge in both**

**aquifers being 25-30% of the average precipitation in the region (c. 440 mm/year) (Martos-Rosillo et al., 2015)"**

**Martos-Rosillo, S., González-Ramón, A., Jiménez-Gavilán, P., Andreo, B., Durán, J. J., & Mancera, E. (2015). Review on groundwater recharge in carbonate aquifers from SW Mediterranean (Betic Cordillera, S Spain). *Environmental Earth Sciences*, 74, 7571-7581.**

For the lake level calculations, the authors considered that water inputs only come from the watershed (Line 188), which is assumed to have remained constant over the past 35 kyrs. What about changes in the groundwater fluxes to the basin from the surrounding aquifers during the last 35 kyrs?.

*Answer*: **As we mentioned in the previous comment, for our calculations, groundwater runoff is included in Basin discharge (BD). See how we simplified the water balance from equation 1 to equation 2 in the Methodology section. Moreover, as the climatic conditions do not change significantly during the last 35 kyrs, the FdP playa-lake probably behaved as a discharge system since its inception and groundwater fluxes to the basin probably remained constant. We have included: "With these climatic conditions, the FdP playa-lake probably behaved as a discharge system since its inception." to make it clearer.**

One of the main aims of the manuscript is to analyze the role of climate variability and tectonics activity in lake level changes, but the argument could be stronger if the data sets were compared in the text and the figures. The results of the hydrological modeling (Fig 7) could be plotted with the FdP lake level reconstructions (Höbig et al., 2016) and with Padul paleoprecipitation (Camuera et al., 2022). How the ca 32, 30, 20, 9.8-7.5, 6-5 and 0.8 - 0.25 ka BP higher lake level periods inferred by the hydrological model correspond to those derived by the lake sequence and the regional humid periods interpreted in Padul?. With caution due to the uncertainties of the age model, Höbig et al. (2016) suggested several phases of flooding at LFP, when lake level was low and the runoff increased (ca. 4.5 ka cal BP, 9 ka cal BP, 14.8 ka cal BP, and 18 ka cal BP) and periods of relatively higher lake level during the LGM, HE1, YD, around 8.2 and 4.2 ka. Camuera et al. (2022) identified several regional humid periods: WMHP-3 (39–29 kyr BP), WMHP-2 (27–18.5 kyr BP; WMHP-2.2 at 27–25 kyr BP and WMHP-2.1 at 23–18.5 kyr) and WMHP-1 (15.5–5 kyr BP). Interestingly, the paleohydrological model for FdP shows higher lake levels at around 30 ka, LGM early to mid Holocene and the last centuries. The FdP reconstruction based on sediment sequence (Höbig et al., 2016) showed also higher lake levels during the LGM and some periods of the Late glacial, but an opposite trend during the Holocene with relatively lower lake levels during the early and mid Holocene. Annual recharge of the aquifers during winter rainfall could be a stronger

proxy for lake level variability and seasonality of precipitation could play a large role in paleohydrology of FdP. All these coherences and discrepancies should be addressed in the discussion.

*Answer*: **This is a really interesting point and we have included this comparison in figures. Moreover, we made another figure summarizing the FdP evolution at different stages (Fig. 13 in the new version of the MS). We have also included in the discussion the importance of the climatic variability on the FdP hydroperiod: "During some extremely wet periods described by Camuera et al. (2022) such as WMHP-3 (39–29 kyr BP), WMHP-2 (27–18.5 kyr BP; WMHP-2.2 at 27–25 kyr BP and WMHP-2.1 at 23–18.5 kyr) and WMHP-1 (15.5–5 kyr BP), the FdP playa-lake probably enlarged its hydroperiod to more than 80%. Indeed, it could have behaved as a permanent lake during short periods, as it currently occurs with other lakes in this area (e.g. Amarga Lake; Jiménez-Bonilla et al., 2023)." Comparing with the lake level reconstruction of Höbig et al., 2016 we included in the discussion: "Comparing our lake level reconstruction with that one of Höbig et al. (2016), based on sediment sequence, we observe that the lake level is higher during the LGM and some periods of the Late glacial, but an opposite trend during the Holocene with relatively lower for the Höbig et al. (2016) reconstruction (Fig. 7). It could be due to a recharge to the karstic aquifers during the winter."**

I find a very suggestive outcome of the model that lake level stabilized during wet periods at 5 m higher than current level. The model considered extremely rainy periods of 50 yrs (P =700 mm, ETP = 850 mm and runoff = 75 mm; Fig. 7B) and the water level stabilizes when water inputs are equal to water outputs. Could higher groundwater inputs during humid periods have increased the water balance and allowed higher lake levels?.

*Answer*: **This is an interesting issue. Our model does not contemplate the case of extremely wet periods (Fig. 9), when the water level could be much higher than nowadays. However, during the last decades, which include wet periods with P > 1000 mm/yr, the piezometric level remained stable (Rodríguez-Rodríguez et al., 2016). In the case of greater increase of groundwater level, the AFS would increase because of a water supply from the aquifer, the W/AFS relationship would reduce and then, the equilibrium would be reached before 17 years, but at the same lake level: 5 m. We have included this question in the new version of the MS: "This lake level could be reached before 14 years in case of groundwater level increases." And in the discussion: "During these extremely wet periods, our model does not contemplate a possible water supply from the aquifer, which would reduce the time to reach the maximum water level."**

Other factors that could change the outcome of the model are related to the changes in watershed (W) versus lake basin (AFS) surface areas. Although it is clear the importance of the flat bathymetry of the FdP playa-lake as the W/AFS relationship decreases when the FdP water level increases. But as I understand it that is based on the assumption that watershed surface area is in equilibrium with average flooded surface and that seems counter intuitive with large tectonic changes in the basin. A premise of the model is that W/AFS is constant, and at the same time, areas as La Nava Playa lake were formerly part of the FdP playa lake and had been subsequently separated by the uplift of La Nava fault footwall. Are there any data or hypothesis about when did La Nava subbasin surface drainage was individualized?; Are FdP and La Nava still connected via groundwater flows? .

*Answer*: **In the MS we suggest that the FdP watershed's size didn't change much during its evolution (See: "Some plausible increases/decrease on the FdP watershed (<10 km$^2$, 6% of the current watershed) would not imply significant increases on the W/AFS relationship (Fig. 12)." In the previous MS). Neotectonics partially affected the FdP watershed configuration but did not the area, which mostly remained constant during the past 35 Kyrs. Active faults mostly affected the FdP playa-lake (its flooded area). As reviewer #1 stated, there were a misleading of the concept of FdP watershed and FdP playa-lake that has been solved in the new version of the MS. Moreover, in order to make it clearer, we have included that: "Although the watershed geometry could have changed, the FdP watershed area probably remained constant."**

**La Nava playa-lake is considered as part of the AFS before dried up. It currently behaves as a recharge playa-lake, then it forms part of the FdP watershed. We have included in the current MS that: "Consequently, La Nava playa-lake is currently a recharge system linked to the FdP playa-lake."**

Another line of evidence for changes in the FdP flooded area is the occurrence of a transition zone between fresher and more saline water at 1-2 m deep. But how the 1-2 m deep transition zone could be stable for thousands of years?.

*Answer*: **We acknowledge the reviewer's question. It is true that salt lake systems with large variations in chemical and stable isotopic composition and complex flow systems are not yet well understood. In addition, hydrogeochemical modelling approaches regarding brine evolution have also been poorly addressed in the literature. Anyhow, there are a few studies made in FdP playa lake, mainly from the last decade, that focused in this matter (e.g. Montalban et al., 2017; Heredia et al., 2010; Kohfahl et al., 2008). Tritium analysis of groundwater yielded high concentrations within the Miocene aquifer indicating ground-water ages**

younger than 50 years which points to comparatively fast ground-water flow at a greater distance from the lake. By contrast, brine samples taken from the lake sediments yielded tritium ages of more than 50 years and indicate longer residence times due to low hydraulic conductivities of lacustrine sediments (Rodríguez-Rodríguez et al., 2005). This is a key aspect of FdF hydrogeological system. Extremely low hydraulic conductivities below the lake implies very slow groundwater fluxes. On the other hand, electric tomography profiles for the location and characterization of the brines below and around the playa lake carried out by Ruiz et al. (2006) yielded elevated conductivities reflecting the presence of salt or saltwater up to a maximum depth of 100 m in the northern part below the centre of the lake. Mapping of the uppermost meters in the basin north of the lake shore indicates that the former lake extended about 3 km further north (Kohfahl et al., 2008). So the salt brine below the playa lake and consequently the interface with fresh groundwater is expected to remain stable during thousand years, even if the playa-lake displace its depocenter, as is the case in FdP. We included it in the new version of the MS: "…and it remains during thousand years once a playa-lake dries up".

Heredia, J., García de Domingo, A., Ruiz, J. M., & Araguás, L. (2010). Fuente de Piedra Lagoon (Málaga, Spain): a deep Karstic flow discharge point of a regional hydrogeological system. Advances in research in Karst media, 231-236.

Kohfahl, C., Rodriguez, M., Fenk, C., Menz, C., Benavente, J., Hubberten, H., … & Pekdeger, A. (2008). Characterising flow regime and interrelation between surface-water and ground-water in the Fuente de Piedra salt lake basin by means of stable isotopes, hydrogeochemical and hydraulic data. Journal of Hydrology, 351(1-2), 170-187.

Montalván, F. J., Heredia, J., Ruiz, J. M., Pardo-Igúzquiza, E., de Domingo, A. G., & Elorza, F. J. (2017). Hydrochemical and isotopes studies in a hypersaline wetland to define the hydrogeological conceptual model: Fuente de Piedra Lake (Malaga, Spain). Science of the Total Environment, 576, 335-346.

Rodríguez-Rodríguez, M., Benavente, J., & Moral, F. (2005). High density ground-water flow, major-ion chemistry and field experiments in a closed basin: Fuente de Piedra Playa Lake (Spain). American Journal of Environmental Sciences, 1(2), 164-171.

Ruiz, J. M., Rubio, F. M., Ibarra, P., García de Domingo, A., Heredia, J., & Araguas, L. (2006). Contribución de la tomografía eléctrica en la caracterización del sistema hidrogeológico de la laguna de Fuente de Piedra (Málaga)[Contribution of electrical tomography in the

*characterization of the hydrologic system of the lagoon of Fuente de Piedra (Málaga)]. Las aguas subterráneas en los países mediterráneos, 1, 353-358.*

**Tectonics.**

*Answer*: **Active tectonics plays a crucial role on the FdP playa-lake evolution, then we tried to improve this relevant issue.**

One of the aims of the paper is to examine the role of Quaternary activity of active faults. However, there is no direct information of evidence for Late Pleistocene or Holocene faults activity (La Nava and Las Latas).

*Answer*: **Both Las Latas and La Nava faults affect Pliocene sediments (see results section). Moreover, earthquakes in this area are common, since it is close to these fault zones. The focal mechanisms of these earthquakes are also compatible with their kinematics; then we discuss that both faults are probably active during the Quaternary. In addition, the fault kinematics are compatible with the FdP playa-lake evolution. Now, we have discussed in more detail the age of both faults in the discussion section. We have added an additional reference that deals with this issue: "Indeed, both Las Latas and La Nava faults affect Pliocene sediments. The location of earthquakes and focal mechanisms are compatible with their kinematics (Jiménez-Bonilla et al., 2024). Consequently, both faults were probably active during the Quaternary."**

Although there are several references to tectonic activity of the Humilladero transversal zone during the last 30 ka, it would be better to include a paragraph stating what it is known about recent (< 30 ka) activity. The La Nava and Las Latas are active structures and there are references (Jiménez Bonilla et al,. 2015, 2023) to recent earthquakes, and geomorphological changes. Are there any dating of major activity related to FdP basin inception (Line 425) or evolution?. A general tectonic framework should also be included in a final figure comparing the lake level reconstruction based on the hydrological model, the sediment sequences, and local paleoclimate reconstruction.

*Answer*: **Unfortunately, there is not information about the tectonic activity of the Humilladero transverse zone nor the La Nava and Las Latas faults. This is an interesting topic that may be investigated in the future. In the last figure, we have included the plausible tectonic activity of these faults and their role on the FdP playa-lake evolution.**

**Depositional evolution**

The age models of both lake sequences are of low resolution and with large uncertainties. The long, littoral core has many reversals, and many dates are too old for their stratigraphic location. In the central core, sediments at about 0.7 m below the playa-lake floor are 34±1.5 kyrs, the age of the sediments in the central basin is older than 48 kyrs and the bottom sediments could be as old as 98 kyrs (2012-PL1; Figs. 3A and 4B), but the age of the upper 70 cm is unknown. Both age models do not preclude the presence of erosive hiati. These are the best available age models for the sequences, but the reconstructed lake level time series should be taken with caution, and the comparison with the hydrological model output has to include the uncertainties. Could these large changes in sedimentation rates and/or depositional evolution between the southern, central and northern areas suggest that the FdP basin is actually composed of "independent" subbasins with varied depositional, hydrological and tectonic evolution?. If so, could the northern subbasin have had a different paleohydrological and lake level evolution during the Pleistocene, accounting for the lake deposits at 15 m above lake level?. May be the available data do not allow to discard some of these hypotheses, but it would be worth discussing them in the manuscript.

*Answer*: **This is an interesting question, and the FdP playa-lake could have splitted into separate flooded areas during its evolution at short periods of time. However, the presence of a flat topography at the main current FdP playa-lake and others outcrops that we consider part of the old FdP playa-lake makes it difficult: The La Nava playa-lake shows a really flat surface, but uplifted. The NE edge of FdP playa-lake also shows a flat topography but tilted towards the S and connected with the main FdP playa-lake body without any umbral. Nevertheless, we cannot completely discard the possibility of the formation of subbasins during the FdP evolution, then we added in the new version of the manuscript: "Alternatively, it can not be ruled out that the FdP playa-lake splitted in some subbasins during its evolution. It could also explain the little resemblance between 2012PL1 and 2012-PL2 (Figs. 5 and 6).".**

The lake level evolution is mostly based on gypsum crystal morphology of the long core (Line 310 and Figure 6). "Detrital gypsum" is a common occurrence in shallow playa lakes affected by wind erosion and frequent changes in lake level. The source area of the detrital gypsum in the southern areas could have been the same southern areas and also be the central zone. Are there any possible inferences of lake level evolution for the central core using the same criteria?. Would the 20-120 cm interval with carbonate muds and abundant prismatic gypsum crystals represent an older period of relatively higher lake levels?. What would be the estimated age of this humid period?, 40 – 60 ka ?(Fig 6) (Obert et al., 2022) ?.

*Answer*: **Thank you because this issue makes us to think more about the possible first stages of the FdP playa-lake. As we mentioned in the discussion, the FdP playa-lake inception was probably earlier than 35 kyrs, but we only focused on the last 35 kyrs. The existence of an older wet period could explain some parts of the described sedimentary sequences.**

There is no further information and discussion about the lacustrine terraces found by Höbig et al. (2016) at about 15 m above current lake level at the NE margin. The lacustrine terraces occurring about 10 m about lake level in the southern margin are re-interpreted as Pliocene sandstone (Line 395). As this is a significant piece of evidence for paleohydrological reconstructions, I would expand this section, with some more detailed sedimentological, compositional and depositional information.

*Answer*: **To the NE margin, we interpreted the "lake terraces" of Höbig et al. (2016) as "lake sediments", included in the Fig. 3 and in our results section. To make it clearer, we have better explained it in the discussion including: "In contrast, to the NE of the current FdP playa-lake, these deposits are dark soils with a flat topography which correspond with old lake deposits (Fig. 3)." In contrast, the sediments of the S margin of the playa-lake are not lake deposits, playa-lakes are not high energy environments to generate conglomerates nor sandstones within its sedimentary register. In the S margin, we only observed sandstones and conglomerates whose clasts come from more than 50 km far the playa-lake, out of the current FdP endoreic watershed. Consequently, they are not lake deposits. We have explained it better in the current MS version adding: "Moreover, these conglomerates and sandstones are made up of clasts from the Alboran domain, which is currently out of the FdP endoreic watershed. Then, these sediments were deposited before the FdP watershed inception."**

The title reflect the main aims of the paper but it could be rephrased as "The role of Neotectonics and climate variability on Late-Quaternary hydrological evolution of Fuente de Piedra playa-lake (southern Iberia)".

*Answer*: **We changed the title of the MS into: "The role of neotectonics and climate variability on the Pleistocene-Holocene hydrological evolution of the Fuente de Piedra playa-lake (southern Iberia)" due to the importance of neotectonics in the discussion.**

Please let us know if any more corrections or changes are necessary.

Yours sincerely,

Dr. Alejandro Jiménez Bonilla on behalf of the co-authors

---

## Referee Report (RR1)

Thank you for the opportunity to review the revised version of this manuscript. I am pleased to see that the authors thoughtfully and thoroughly addressed all points raised in my initial review. The revisions have significantly improved the manuscript, both in clarity and scientific rigor, enhancing its value to the journal and its readership.

The authors have effectively addressed my suggestions regarding the discussion of climatic and hydrological forces for the study area. This allows for clearer insights into the controlling factors of the playa-lake evolution, making the findings more accessible and compelling for readers. The adjustments to the figures are especially valuable and significantly improved their readability. Two small things caught my eye:

- The labelling in Fig. 10 should probably read "Lake terraces 10 m above the lakebed" and, similarly "Lake deposits 15 above the lakebed".
- In the newly introduced summary figure (Fig. 12), the La Nava playa-lake shows some sinistral displacement, but the corresponding La Nava fault is labelled with a dextral sense of shear. A sinistral sense of shear of the La Nava fault is indicated in the text and in the other figures. Also, there are small green checkmarks along the shoreline which I do not understand, and the red "E" labels probably refer to evaporite formation.

I also appreciate the authors' attention to enhancing the clarity of the manuscript's structure and flow, which now guides readers more intuitively through the research process and findings.

In summary, I find the manuscript to be scientifically sound and well-presented in its current form. I am confident that it will make a valuable contribution to the field and believe it is now suitable for publication.

Thank you once again for the opportunity to review this work.

---

## Author Response (AR2)

HESS

30, October 2024

Dear Editor:

Please convey our gratitude to the reviewers for their detailed reports, which will considerably improve the manuscript.

**Regarding comments from Reviewer #1 Martin Reiser**

**Specific comments**

The labelling in Fig. 10 should probably read "Lake terraces 10 m above the lakebed" and, similarly "Lake deposits 15 above the lakebed".

*Answer*: **We appreciate the reviewer´s comment and agree with that. We changed "upper" into "above" in Fig. 10.**

In the newly introduced summary figure (Fig. 12), the La Nava playa-lake shows some sinistral displacement, but the corresponding La Nava fault is labelled with a dextral sense of shear. A sinistral sense of shear of the La Nava fault is indicated in the text and in the other figures. Also, there are small green checkmarks along the shoreline which I do not understand, and the red "E" labels probably refer to evaporite formation.

*Answer*: **We thanks the reviewer for this comment. Indeed, La Nava fault is a sinistral sense shear zone, then we changed it in the Fig. 12. Green checkmarks at the shoreline try to simulate the vegetation, but we removed it in order to simplify the figure. "E" means evaporation, but it is not needed in the figure, consequently we also removed it from the Fig. 12.**

Please let us know if any more corrections or changes are necessary.

Yours sincerely,

Dr. Alejandro Jiménez Bonilla on behalf of the co-authors